

# Semivolatile POA and parameterized total combustion SOA in CMAQv5.2: impacts on source strength and partitioning

Benjamin N. Murphy[1], Matthew C. Woody[1], Jose L. Jimenez[2,3], Ann Marie G. Carlton[4], Patrick L. Hayes[5], Shang Liu[2], Nga L. Ng[6,7], Lynn M. Russell[8], Ari Setyan[9], Lu Xu[6], Jeff Young[1], Rahul A. Zaveri[10], Qi Zhang[11], and Havala O. T. Pye[1]

[1]National Exposure Research Laboratory, U.S. Environmental Protection Agency, Research Triangle Park, NC, USA
[2]Cooperative Institute for Research in Environmental Sciences, University of Colorado, Boulder, CO, USA
[3]Department of Chemistry and Biochemistry, University of Colorado, Boulder, CO, USA
[4]Department of Chemistry, University of California, Irvine, CA 92697, USA
[5]Department of Chemistry, Université de Montréal, Montréal, QC, Canada
[6]School of Chemical and Biomolecular Engineering, Georgia Institute of Technology, Atlanta, GA, USA
[7]School of Earth and Atmospheric Sciences, Georgia Institute of Technology, Atlanta, GA, USA
[8]Scripps Institution of Oceanography, University of California, San Diego, La Jolla, California, USA
[9]EMPA, Swiss Federal Laboratories for Materials Science and Technology, 8600 Dübendorf, Switzerland
[10]Atmospheric Sciences and Global Change Division, Pacific Northwest National Laboratory, Richland, WA, USA
[11]Department of Environmental Toxicology, University of California, Davis, California, USA

*Correspondence to:* Benjamin N. Murphy (murphy.benjamin@epa.gov)

**Abstract.** Mounting evidence from field and laboratory observations coupled with atmospheric model analyses show that primary combustion emissions of organic compounds dynamically partition between the vapor and particulate phases, especially as near-source emissions dilute and cool to ambient conditions. The most recent version of the Community Multiscale Air Quality (CMAQ) Model v5.2 accounts for the semivolatile partitioning and gas-phase aging of these primary organic aerosol (POA) compounds consistent with experimentally derived parameterizations. We also include a new surrogate species, potential secondary organic aerosol from combustion emissions (pcSOA), which provides a representation of the SOA from anthropogenic combustion sources that could be missing from current chemical transport model predictions. The reasons for this missing mass likely include the following: 1) unspeciated semivolatile and intermediate volatility organic compound (SVOC and IVOC, respectively) emissions missing from current inventories, 2) multigenerational aging of organic vapor products from known SOA precursors (e.g. toluene, alkanes, etc), 3) underestimation of SOA yields due to vapor wall losses in smog chamber experiments, and 4) reversible organic-water interactions and/or aqueous-phase processing of known organic vapor emissions. CMAQ predicts the spatially-averaged contribution of pcSOA to OA surface concentrations in the continental United States to be 38.6% and 23.6% in the 2011 winter and summer, respectively.

Whereas many past modeling studies focused on a particular measurement campaign, season, location, or model configuration, we endeavor to evaluate the model and important uncertain parameters with a comprehensive set of United States-based model runs using multiple horizontal scales (4 km and 12 km), gas-phase chemical mechanisms, seasons and years. The model with representation of semivolatile POA improves predictions of hourly OA observations over the traditional nonvolatile model at sites during field campaigns in southern California (CalNex, May-June 2010), northern California (CARES, June 2010), the



southeast US (SOAS, June 2013; SEARCH, January and July, 2011). Model improvements manifest better correlations (e.g. correlation coefficient at Pasadena at night increases from 0.38 to 0.62) and reductions in underprediction during the photochemically active afternoon period (e.g. bias at Pasadena from -5.62 to -2.42 $\mu g\,m^{-3}$). Daily-averaged predictions of observations at routine monitoring networks from simulations over the continental U.S. (CONUS) in 2011 show modest improvement

during winter with mean biases reducing from 1.14 to 0.73 $\mu g\,m^{-3}$, but less change in the summer when the decreases from POA evaporation were similar to the magnitude of added SOA mass. Because the model-performance improvement realized by including the relatively simple pcSOA approach is similar to that of more-complicated parameterizations of OA formation and aging, we recommend caution when applying these more-complicated approaches as they currently rely on numerous uncertain parameters.

The pcSOA parameters optimized for performance at the southern and northern California sites lead to higher OA formation than is observed in the CONUS evaluation. This may be due to any of the following: variations in real pcSOA in different regions or time periods, too high concentrations of other OA sources in the model that are important over the larger domain, or other model issues such as loss processes. This discrepancy is likely regionally and temporally dependent and driven by interferences from factors like varying emissions and chemical regimes.

# 15  1  Introduction

A substantial fraction of atmospheric particles are made of organic compounds (Zhang et al., 2007; Murphy et al., 2006). This is true over the continental United States, where pollutants introduced by human sources often interact in complex ways with compounds from natural sources (Hallquist et al., 2009; De Gouw and Jimenez, 2009). In order to fully describe the impacts that human activities have on public health, regional haze, and climate change via airborne particles, we must be able to

quantify the capacity for relevant organic compounds to form and maintain airborne particulate mass (Carlton et al., 2010). To achieve this end, atmospheric models routinely rely on estimates of volatility (i.e. vapor pressure or saturation concentration) and solubility in a predominantly aqueous phase. Although research efforts demonstrate clearly the importance of organic solubility in water on phase partitioning and particle viscosity (Zhang et al., 2012; Hodas et al., 2015; Pajunoja et al., 2015; Riipinen et al., 2015; Wania et al., 2015; Isaacman-VanWertz et al., 2016; Jathar et al., 2016b; Pye et al., 2017), improvements

to the conceptual model of organic compounds partitioning into an organic-rich particulate phase alone are useful before the entire system is treated holistically.

    It is clear from available experimental data that primary organic aerosol (POA), which is operationally defined as the population of organic compounds that are emitted in the particulate phase, is a more dynamic quantity than originally prescribed in chemical transport models (Lipsky and Robinson, 2006; Donahue et al., 2009; Grieshop et al., 2009; Huffman et al., 2009b, a;

May et al., 2013a, b, c). Not only do these studies show that POA evaporates upon dilution of exhaust from multiple sources and that it is often more volatile than SOA (which had been treated as semivolatile in models), but the semivolatile vapors that are produced are most likely susceptible to continued oxidation in the gas phase. Robinson et al. (2007) first demonstrated, by applying a chemical transport model to predict air quality over the eastern U.S. in July 2001, that treating primary par-



ticulate compounds as semivolatile improves model performance for bulk organic aerosol (OA) mass predictions and better predicts the relative importance of POA and secondary organic aerosol (SOA), which is formed from the oxidation products of volatile organic compound (VOC) gases. Moreover, Robinson et al. (2007) predicted that compounds with intermediate volatility (IVOCs), that is, volatility higher than that of semivolatile organic compounds (SVOCs) and lower than traditionally

defined VOCs, were emitted in significant quantities and that their oxidation products could condense and form OA downwind of combustion sources. An IVOC mass emission scale factor of 1.5 times the POA + SVOC emission rate led to significant contribution of OA derived from IVOCs when applied in their model. However, that scale factor was uncertain, as was the overall efficiency with which those IVOCs formed SOA (Shrivastava et al., 2008).

More recently, both experimental and model studies have sought to constrain combustion-related SVOC and IVOC emissions

as well as their subsequent SOA yields (Tkacik et al., 2012; Jathar et al., 2014; Zhao et al., 2014, 2016b). Progress has been made characterizing the overall significance of this OA production pathway, but the variability among vehicles and other combustion sources is uncertain, as are the effects of operating (e.g. load, burn temperature, etc) and environmental conditions. Additional limitations exist in our understanding of other important processes for SOA formation. First, multigenerational aging of anthropogenic and biogenic condensable organic vapors could affect OA partitioning if chamber experiments do not

proceed for a long enough duration to fully capture the processes, and these processes lead to a net increase or decrease in the volatility of the population (Hallquist et al., 2009; Cappa and Wilson, 2012; Donahue et al., 2013; Hildebrandt Ruiz et al., 2015; Jathar et al., 2016a). Second, it is likely that previous estimates of SOA yields from well-known VOC precursors (e.g. toluene, xylene, alkanes, isoprene, terpenes, etc) derived from smog chamber experiments were biased by unaccounted loss of semivolatile organic vapors to the chamber walls (Hildebrandt et al., 2009; Loza et al., 2010; Zhang et al., 2014, 2015; Cappa

et al., 2016; Nah et al., 2016; Krechmer et al., 2016; Saha and Grieshop, 2016; Nah et al., 2017). The magnitude of this bias likely varies substantially among experimental systems (i.e. precursor and oxidant identity) and experimental conditions (e.g. seed condentration, relative humidity, temperature, etc) (Krechmer et al., 2016; Nah et al., 2016). Finally, aerosol liquid water may absorb soluble organic vapors and shift the partitioning to enhance OA concentrations as explored by Pye et al. (2017).

Several models have now incorporated the semivolatile POA partitioning phenomena demonstrated by Robinson et al.

(2007), relying on a framework popularly referred to as the Volatility Basis Set (VBS) (Donahue et al., 2006, 2011a, b). Examples of these models include box- (Dzepina et al., 2009, 2011; Hayes et al., 2015; Ma et al., 2016), regional- (Lane et al., 2008; Shrivastava et al., 2008; Murphy and Pandis, 2009; Tsimpidi et al., 2010; Hodzic et al., 2010; Hodzic and Jimenez, 2011; Murphy et al., 2011; Ahmadov et al., 2012; Bergstrom et al., 2012; Zhang et al., 2013; Koo et al., 2014; Matsui et al., 2014; Knote et al., 2015; Tuccella et al., 2015; Zhao et al., 2016a) as well as global-scale (Pye and Seinfeld, 2010; Farina et al., 2010;

Jathar et al., 2011; Jo et al., 2013; Tsimpidi et al., 2014; Tsigaridis et al., 2014) applications. Each implementation has made assumptions about the emissions and SOA yields of IVOCs, the reaction rates and products of multigenerational aging, and in some cases, the biases from chamber wall losses of semivolatile vapors. Despite wide application of the VBS framework, large-scale modeling studies, which include many other uncertainties from e.g. emissions inventories, scavenging processes and oxidant concentrations, have made slow progress towards reducing the uncertainty of the many free parameters related to

OA emission, formation and processing. However, the details of the implementation in CTMs affect OA source apportionment



(Hodzic et al., 2010; Pye and Seinfeld, 2010; Murphy et al., 2012; Bergstrom et al., 2012; Matsui et al., 2014; Hodzic et al., 2014). Current implementations in global models result in a spread of 1-2 orders of magnitude in OA concentrations across models, especially aloft and in remote areas (Tsigaridis et al., 2014). Therefore, caution should be exercised when adopting an SOA formation mechanism that appears detailed but is largely unconstrained.

Hodzic and Jimenez (2011) showed that an optimized two-parameter fit for SOA formation from anthropogenic sources (constraining the amount and timescale of SOA formation) could successfully capture trends observed in multiple field studies (DeCarlo et al., 2010). That parameterization reasonably reproduced AMS observations within and downwind of a megacity. The simple SIMPLE (SIMPLifiEd parameterization of combustion SOA) demonstrated by Hodzic and Jimenez (2011) involved one VOC surrogate which, when oxidized by hydroxyl radicals, formed one condensable vapor that partitioned irreversibly to
the particle phase. The use of non-volatile SOA as an approximation was based on the observation that real SOA had low volatility, and generally much lower than used in prior models (Dzepina et al., 2009; Huffman et al., 2009b). The uncertain parameters, the emission of the VOC and its oxidation reaction rate, were optimized for performance in the regional model CHIMERE. This method was reapplied to analyze urban-scale data at Pasadena (Hayes et al., 2015). Most recently, Woody et al. (2016) applied the SIMPLE method in CMAQ to Pasadena and showed substantially improved model-measurement
agreement, while Kim et al. (2015) applied it to the SE U.S. and showed agreement with fossil/non-fossil carbon observations.

We document here the implementation and evaluation of semivolatile partitioning of POA and addition of observed, but unspeciated, SOA in the Community Multiscale Air Quality (CMAQ) model version 5.2. The model approach uses the VBS framework to account for the dynamic partitioning and aging of POA emissions, without modifying the two-product approach for SOA formation from traditional SOA precursor VOCs (e.g. toluene, xylene, alkanes, isoprene, terpenes, etc). We further
introduce a new surrogate species, potential SOA from combustion emissions (pcSOA) to account for missing mass from IVOC oxidation, multigenerational aging of (anthropogenic) secondary organic vapors (from IVOC, and VOC precursors), biases in SOA yields from vapor wall losses, and enhanced organic partitioning to the condensed aqueous phase. In addition to these sources, pcSOA could account for mass from oxidation of as-yet unidentified sources of SOA precursors. The contribution of this surrogate species to the total OA burden is governed by two parameters, analogous to the simple method of Hodzic and
Jimenez (2011), the emission rate of the SOA precursor and the precursor reaction rate with hydroxyl radicals. In the following sections we present a comprehensive evaluation of the improvement in CMAQ OA predictions at two horizontal resolutions (4 and 12 km), over multiple seasons and years, and using multiple gas-phase chemical mechanisms. We explore the sensitivity of CMAQ OA to two parameters directly impacting pcSOA formation and recommend important areas of future work.

## 2  Organic aerosol module configuration

In developing CMAQv5.2, we have updated the OA module to be more consistent with current understanding of POA emission and to better represent the magnitude of SOA formation from anthropogenic combustion sources.



## 2.1 POA semivolatile partitioning and aging

The model accounts explicitly for the gas/particle partitioning of primary organic compounds and their multigenerational oxidation products with a range of volatility, similar to the 1.5-D volatility basis set scheme of Koo et al. (2014). Table 1 lists the new species in the aerosol module as well as important properties. Like Koo et al. (2014), we choose surrogates with volatilities that are relevant for typical atmospheric loadings (i.e. 0.1 - 1000 µg m$^{-3}$). Donahue et al. (2011a) related the volatility, expressed in terms of an effective saturation concentration $C^*$, to both carbon number and O:C using structure activity relationships, vapor pressure estimation methods like SIMPOL (Pankow and Asher, 2008) and available vapor pressure measurements, and we apply their methods here (see supporting information). For the directly emitted species (LVPO1, SVPO1, SVPO2, SVPO3 and IVPO1), the carbon number (C17-C19) and O:C (0-0.1) of the new surrogate species are chosen to be consistent with laboratory and field observations of the properties of POA from combustion sources (Aiken et al., 2008; Huffman et al., 2009a; Presto et al., 2012; May et al., 2013a, b, c; Canagaratna et al., 2015). Previously, Simon and Bhave (2012) used a separate tracer, PNCOM, in CMAQ to track the contribution of non-carbon organic matter to the total POA mixture. During the emission input generation procedure, PNCOM emission factors were informed by the OM:OC of individual sources (e.g. mobile, 1.25; cooking, 1.4; biomass burning, 1.7). In the present model we sum the primary organic carbon and non-carbon emission factors in order to preserve the source-aware mass emissions developed by Simon and Bhave (2012). Because the new primary surrogate species have fixed O:C (table 1), the source-resolution in O:C is lost. This limitation can be overcome in future model versions by treating, for example, biomass burning OA independently with additional model species.

For the oxidation products of the directly emitted species (LVOO1, LVOO2, SVOO1, SVOO2, SVOO3), we have chosen higher OM:OC (and thus lower carbon number) values than Koo et al. (2014) to bound the typical observed range of OM:OC reported from ambient studies, including Aiken et al. (2008) and Canagaratna et al. (2015) for multiple U.S. and global sites (1.3-2.25) and Simon et al. (2011) for routine monitoring networks in the continental U.S. (0.79-2.15). The species enthalpy of vaporization is assumed to be a linear function of $C^*$ (May et al., 2013b; Epstein et al., 2010) and the Henry's law coefficient is prescribed at 2 x $10^8$ M atm$^{-1}$, which is in the range reported for oxidized organic vapors by Hodzic et al. (2014) and Pye et al. (2017).

We employ one volatility distribution, from Robinson et al. (2007) for the partitioning calculation of the primary organic emissions from all combustion sources. Koo et al. (2014) found some differences in their results when they used separate distributions for biomass burning, gasoline and diesel sources. The Robinson et al. (2007) distribution results in more aggressive POA evaporation than the source-resolved multi-distribution parameterization. Dzepina et al. (2009) showed that the Robinson et al. (2007) parameterization resulted in POA volatility that was similar to thermodenuder observations. We performed a sensitivity simulation using a single modified distribution calculated by summing the source-resolved distributions weighted by their total emissions in the California domain. This simulation resulted in maximum hourly POA increases of 10-15%. Future work will incorporate emerging measurements of primary organic compound volatility information directly into the generation of emission inventories, along with estimating sensitivities to key factors like ambient temperature, operating conditions, fuel type, etc. We do not apply a scaling factor to the POA input emission factor to, for example, introduce new SVOCs to the model.





Our approach assumes that existing inventory POA emission factors are actually representative of the total gas- plus particle-phase mass with saturation concentration below about 3200 $\mu g\,m^{-3}$. It is unclear to what extent SVOCs are missing from the emission factors that inform current inventories as these experiments were conducted at low dilution ratios and enhanced partitioning to the particulate phase. This issue should be resolved by taking into account temperature and organic aerosol loading in past and future laboratory-scale experiments when that data is available.

Additionally, some of the semivolatile primary particle mass that evaporates condenses back to the particle phase after oxidation. Although direct observations of primary combustion-derived SVOC aging are rare, both field results (Chan et al., 2013) and their relatively large carbon numbers support that they are susceptible to OH oxidation and SOA formation in the atmosphere. We apply an OH oxidation reaction with a rate constant equal to 4.0 x $10^{-11}\ cm^3\ molec^{-1}\ s^{-1}$ to simulate this aging, consistent with previous modeling studies (Grieshop et al., 2009; Farina et al., 2010; Murphy and Pandis, 2009; Koo et al., 2014; Zhao et al., 2016a). The stoichiometric coefficients for this oxidation step are derived using 2D-VBS theory for low-$NO_x$ systems (Donahue et al., 2011a, b) and are documented in supporting information. The branching ratio for functionalization and fragmentation processes is parameterized as in Eq. 1.

$$\beta_{frag} = (O:C)^{0.4} \tag{1}$$

The aging configuration tends to produce lower volatility (thus more particulate) mass than that used by Koo et al. (2014) and produces slightly less total organic mass (OM) than the full 2D-VBS model of Chuang and Donahue (2015) (see supporting information; Fig. S1a). The bulk O:C enhancement due to aging is more vigorous in this model compared to the full 2D-VBS. However, the rate of O:C enhancement is about half that of the POA aging scheme implemented in older CMAQ versions (Simon and Bhave, 2012) (see supporting information; Fig. S1b).

## 2.2 SOA from traditional precursors

SOA formed from oxidation of traditional VOC sources (isoprene, monoterpenes, sesquiterpenes, benzene, toluene, xylene, alkanes, and PAHs) are documented in Carlton et al. (2010) and Pye and Pouliot (2012). The SOA yields of these compounds has been unchanged in the development of CMAQv5.2, although updates to their molecular properties (solubility, molecular weight, OM:OC, etc) were incorporated to maintain consistency throughout the model processes (Pye et al., 2017). CMAQv5.2 also includes aqueous processing of isoprene epoxides, glyoxal and methylglyoxal as well as production of oligomeric species from particle-phase reaction of traditional SOA compounds. We do not employ the recent phase liquid-liquid phase partitioning algorithms implemented by Pye et al. (2017).

## 2.3 Potential SOA from combustion emissions

Because the individual contributions to the total bias from missing IVOC oxidation, missing multigenerational aging of VOC oxidation products, underrepresenting SOA yields because of chamber wall losses and uptake of organics by aerosol liquid water are all uncertain, we introduce one surrogate aerosol species, pcSOA, to address the total missing OA mass. Three essential features are common among the aforementioned uncertain processes: they involve emission of gas-phase compounds, some



degree of photooxidation of those compounds, and condensation of the resulting products. Thus, we simulate this formation process analogously to the SIMPLE approach of Hodzic and Jimenez (2011). A new surrogate VOC species (potential VOC from combustion emissions, pcVOC) is introduced with an emission rate that is scaled to the POA mass emission rate, and is oxidized with OH to form a low volatility condensable vapor, potential secondary organic gas from combustion emissions,

pcSOG (table 2). For the BASE simulation we assume an OH reaction rate constant equal to the optimal rate constant reported by Hayes et al. (2015), and we convert the optimal emission scale factor of Hayes et al. (2015), which is scaled to CO emissions to one that can be applied to POA emissions. To convert from $g^{-1}$ CO to $g^{-1}$ POA, we multiply by the average ratio of CO emissions to POA emissions in the CONUS11 emissions dataset, 0.82. A promising alternative to scaling emissions to POA emissions could be to apply an SOA yield directly to the oxidation of total non-methane organic gases (NMOG), as

demonstrated in Jathar et al. (2014) and implemented by Jathar et al. (2016c) for vehicle emissions. Unfortunately, NMOG emissions inventory data as they currently stand, do not always take into account the evaporation of primary vapors nor are they always treated consistently by typical emissions processing practices. We consider the implementation of semivolatile POA and missing combustion SOA in CMAQv5.2 an important step toward an even more rigorous future treatment, whether that involves scaling emissions to CO or NMOG. As such, we have prioritized minimizing the need for reprocessing emissions

inputs for users of previous CMAQ versions, but this does leave room for future refinement of source-specific SOA formation from unspeciated vapor emissions.

The emissions of pcVOC are applied to all combustion-related area and point sources of primary organic carbon except for wildfires (classified as point sources), as it has been shown that fire plumes do not exhibit an SOA enhancement as strong as those dominated by urban emissions (Cubison et al., 2011). Thus, important sources like onroad vehicles, power plants, and

commercial cooking are all included. Hayes et al. (2015) and Ma et al. (2016) showed that residential as well as commercial cooking sources of SOA are potentially an important contribution to nonfossil carbon in Pasadena, and residential cooking sources are missing from the inventories used here. The inclusion of both residential wood burning and agricultural burning area sources in the pcSOA parameterization should be investigated further and potentially treated separately in the future by adding more detailed on-line source resolution to the CMAQv5.2 OA model. The partitioning between pcSOG and pcSOA is

governed by a saturation concentration equal to $1 \times 10^{-3}$ µg m$^{-3}$. No further reactions are included for either pcSOA or pcSOG. The low volatility of pcSOA makes this OA configuration behave similarly in many ways to the approach of Shrivastava et al. (2013), which attempts to account for semisolid OA behavior and oligomer formation; however there are important differences. All OA mass (including pcSOA) is treated as an absorbing medium in our approach and pcSOA is created through direct oxidation and condensation of gas-phase species, not through a particle-phase transformation. Future work will address the

role of diffusion-limited OA absorptive partitioning and its dependence on relative humidity. The emission scale factor and oxidation rate constant are uncertain parameters and we present a sensitivity analysis of them in order to both demonstrate the relevance of this uncertainty to total OA concentrations and to recommend acceptably-performing parameters for use in operational CMAQv5.2 simulations.



## 3 Model application

We apply CMAQv5.2 with the enhanced OA module to three distinct simulation domains: the continental United States, the eastern United States and California. The time periods and chemistry options, detailed below, are chosen to leverage existing observations while also evaluating the model under various configurations. The individual simulations are summarized in table
5   3.

### 3.1 Observations

In order to connect the performance of the updated model with past CMAQ versions, we evaluate predictions against organic carbon (OC) observations from routine monitoring networks including the Interagency Monitoring of Protected Visual Environments (IMPROVE) and Chemical Speciation Network (CSN) datasets. We note that IMPROVE data had a 27% low bias
relative to collocated SEARCH observations during summer 2013 in the SE U.S., which is thought to be due to evaporation during sampling and transport (Kim et al., 2015). That bias was not observed for non-summer samples. Additionally, we focus on three intensive measurement campaigns: the California Research at the Nexus of Air Quality and Climate Change (CalNex), the Carbonaceous Aerosols and Radiative Effects Study (CARES), and the Southern Oxidant and Aerosol Study (SOAS). At these sites, CMAQ OA predictions are evaluated against High-Resolution Time-of-Flight Aerosol Mass Spectrometers (HR-
ToF-AMS) data. The Southeastern Aerosol Research Characterization (SEARCH) network sites located in the southeast U.S. provide semi-continuous OC observations also used to evaluate model capability during the SOAS time period. Two urban sites, Jefferson Street, Atlanta, GA and Birmingham, AL, and one rural site, Yorkville, GA are selected. During the CARES and SOAS campaigns, thermodenuder measurements allow us to make a quantitative evaluation of the OA volatility distribution predicted by the model.
The CalNex campaign characterized atmospheric composition at two sites in Southern California, Pasadena and Bakersfield, from 15 May to 29 June 2010 (Ryerson et al., 2013). The Pasadena site was located 18 km northeast and generally downwind of downtown Los Angeles, while the Bakersfield site was situated at the southern end of the San Joaquin Valley (SJV). Several studies characterized OA properties and likely sources at Pasadena and the larger LA basin with observations (Washenfelder et al., 2011; Bahreini et al., 2012; Borbon et al., 2013; Hersey et al., 2013; Liu et al., 2012a; Hayes et al., 2013; Fast et al., 2014;
Knote et al., 2014; Ma et al., 2016) while others combined measurements with models at multiple scales (Chan et al., 2013; Ensberg et al., 2014; Hayes et al., 2015; Baker et al., 2015; Woody et al., 2016). On average, organic compounds comprised 41% of the total $PM_1$ mass concentrations at Pasadena, and exhibited a strong diurnal-averaged peak concentration in the mid-afternoon (3 pm) (Hayes et al., 2013). At Bakersfield, OA compounds comprised 56% of the total $PM_1$ mass, and fossil fuel sources were determined to contribute 80-90% of those components (Liu et al., 2012b; Gentner et al., 2014). Average O:C
at Pasadena ranged from about 0.38-0.48 throughout the campaign with some extreme values extending to 0.8.

The CARES campaign took place in Northern California from June 2 to 28, 2010 and conducted measurements at two ground sites, Sacramento and Cool, and from the DOE G-1 aircraft (Zaveri et al., 2012). The Cool site is a suitable location to observe interactions among anthropogenic and biogenic pollutants due to its proximity to outflow from Sacramento and the





forested foothills of the Sierra Nevada Mountains (Setyan et al., 2012, 2014). Like those at the CalNex stations, observations during CARES indicated that organic compounds dominated (~80%) the particle phase mass composition at both sites (Setyan et al., 2012; Shilling et al., 2013). Shilling et al. (2013) and Setyan et al. (2012) showed that the combination of anthropogenic and biogenic pollutants in the vicinity of Sacramento could enhance SOA formation significantly beyond instances when just

anthropogenic or just biogenic sources contributed.

The SOAS campaign also targeted observations of anthropogenic-biogenic interactions, but at three sites (Centreville, AL; Birmingham, AL; and Look Rock, TN) in the substantially differing southeastern U.S. environment during June and July 2013 (Carlton et al., Submitted, Aug 2016). Anthropogenic emissions, including $NO_x$ and $SO_2$, have large influence on isoprene and monoterpene SOA in both rural and urban platforms (Budisulistiorini et al., 2015; Hu et al., 2015; Xu et al., 2015; Rattanavaraha

et al., 2016). Pye et al. (2015) and Xu et al. (2015) found that organic nitrate chemistry likely played a substantial role in the formation of observed LO-OOA compounds at the Centerville site. Xu et al. (2015) also found that SOA from isoprene oxidation contributed 18% of the total OA averaged throughout the campaign.

## 3.2 Model configuration and analysis

We configured CMAQv5.2 to simulate air quality over California (CAL) at a horizontal resolution of 4 km by 4 km during

May and June 2010. The input parameters are documented extensively by Baker et al. (2013), Baker et al. (2015) and Woody et al. (2016) and are summarized briefly here. Meteorological inputs are generated using the Weather Research and Forecasting (WRF) model, Advanced Research WRF (ARW) core version 3.1 (Skamarock et al., 2008). Baker et al. (2013) evaluated the meteorological parameters (temperature, wind speed, wind direction, etc) and found acceptable performance. The boundary conditions were generated from running a coarser resolution CMAQ simulation from December 2009 through June 2010.

That coarse simulation used boundaries driven by a GEOS-Chem (v8-03-02) simulation from the same period (Henderson et al., 2014). Electrical generating unit and other point source emissions were estimated specifically for 2010 using continuous emissions monitoring data. Mobile source emissions were generated using a combined approach of generation by SMOKE-MOVES integration platform (US Environmental Protection Agency, 2014) and projection by the California Air Resources Board. Other anthropogenic emissions are based on the 2011 National Emissions Inventory (NEI) version 1 (US Environmental

Protection Agency, 2014). Day-specific fire emissions are incorporated into the model inputs, although they do not impact concentrations in Pasadena to a large extent (Hayes et al., 2013). Setyan et al. (2012) and Shilling et al. (2013) also found low impact from biomass burning at both CARES sites. Residential meat-cooking emissions are not included in the emissions inventory as pointed out by Woody et al. (2016). That study found that CMAQ underpredicted cooking-influenced OA by a factor of 2-4, depending on the time of day and assumptions about the volatility of meat cooking emissions. Biogenic vapor

emissions are generated by the Biogenic Emission Inventory System (BEIS) v3.14 (Carlton and Baker, 2011). A modified version of Carbon Bond 5 (CB05e51) simulated gas-phase chemical kinetics including ozone formation/destruction and OA formation among other processes.

The SOAS campaign was addressed with a second simulation over the eastern United States (EUS) during June 2013 at 12 km by 12 km horizontal resolution. The WRF core model version 3.6.1 was applied for meteorological inputs, and boundary



conditions were obtained from a 36 km by 36 km CMAQ simulation using boundary conditions informed by GEOS-Chem (Henderson et al., 2014). As explained by Pye et al. (2015) and Pye et al. (2017), anthropogenic emissions were based on the EPA National Emission Inventory (NEI) 2011 v1, and biogenic emissions were obtained from the Biogenic Emission Inventory System (BEIS) v3.6.1. This implementation included enhanced detail in the formation of isoprene SOA and organic nitrate

kinetics/partitioning. Fire emissions were based on the latest version of the Satellite Mapping Automated Reanalysis Tool for Fire Incident Reconciliation (SMARTFIRE) system (www.airfire.org/smartfire/). The Statewide Air Pollution SAPRC07 mechanism was used to predict gas-phase chemistry.

In order to extend results further, we perform a suite of simulations over the continental United States at 12 km by 12 km horizontal resolution for the entire year 2011 (CONUS11), thereby probing seasonal differences in model performance.

Year-to-year variability is assessed with additional runs over the continental U.S. during January and July 2002 (CONUS02). Anthropogenic, biogenic and day-specific fire emissions for the CONUS simulations are documented by Appel et al. (2016). The CONUS02 meteorological inputs were generated by WRF version 3.1 and anthropogenic emissions were derived from the 2002 NEI (Bash et al., 2013). The OA boundary conditions are downscaled from GEOS-Chem simulations and we prescribe an OM:OC for these compounds equal to 2.0. We estimated organic carbon (OC), which was measured at each site, from the

model output by taking into account the species-dependent O:C and calculating an OM:OC using the approach of Simon and Bhave (2012).

To probe sensitivity to the uncertain pcVOC emission scale factor and oxidation rate constant, we explore a series of perturbation runs varying both of these parameters (table 4). The parameter combinations are chosen to bound acceptably performing estimates of the parameters at Pasadena (Hayes et al., 2015). We convert the perturbed pcVOC emissions scale factors from

$g^{-1}$ CO to $g^{-1}$ POA with the same conversion factor used for the BASE simulation. In addition to these sensitivity cases, we run a case that uses the same OA module configuration as CMAQv5.1 (nvPOA). Specifically, the nvPOA case includes nonvolatile POA, POA aging consistent with Simon and Bhave (2012) and SOA formation from traditional VOC precursors. No pcSOA is implemented for the nvPOA case . We apply all of these cases to the CAL domain and to both January and July of the CONUS11 domain. Meanwhile, due to computation time constraints, we only examine the BASE case, best-performing

CONUS11 case (low-emission-base-reaction; LEBR), and the nvPOA case for the other months of 2011 and for the EUS and CONUS02 domains.

For semi-quantitative evaluations generally within individual or among similar datasets, we utilize standard statistical metrics to characterize model performance. These metrics include e.g. mean bias (MB; μg m$^{-3}$), mean normalized error (MNE) and correlation coefficient (r). For more rigorous quantitative analysis, we employ the normalized mean bias factor (NMBF, Eq. 2)

and normalized mean absolute error factor (NMEF, Eq. 3) developed by Yu et al. (2006).

$$NMBF = S\left[\exp\left(\left|\ln\frac{\sum M_i}{\sum O_i}\right|\right)\right] - 1 \tag{2}$$

$$NMEF = \frac{\sum|M_i - O_i|}{(\sum O_i)^{[1+S]/2}(\sum M_i)^{[1-S]/2}} \tag{3}$$



where M are model predictions, O are observations and S = $(\bar{M} - \bar{O})/|\bar{M} - \bar{O}|$. These metrics were specifically designed to address biases that arise from comparing datasets with substantially different mean values (e.g. urban and rural locations) and widely varying degrees of over- and under-prediction. If NMBF is positive (e.g. 0.6), then the model overestimates the observations by a factor of 1 + NMBF (1.6), on average. If NMBF is negative (e.g. -0.6), the model underestimates the

observations by a factor of 1 - NMBF (1.6), on average. The NMEF is interpreted as follows; if NMEF = 1.8, then the model gross error is 1.8 times the mean observation for overprediction or 1.8 times the mean model prediction for underprediction, on average.

## 4   Results and discussion

### 4.1   Evaluation against continuous OA mass observations

CMAQv5.2 with semivolatile POA and pcSOA (BASE) successfully captured the day/night OA variability substantially better than the model with nonvolatile POA (nvPOA) (Fig. 1). Across all sites, improved predictions were due directly to both the volatilization of primary emissions in the morning and night, and the increased role of secondary formation via pcSOA in the afternoon. We expect these two phenomena to play the largest role in urban locations like Los Angeles, with high combustion emissions and active photochemistry. At Pasadena, the daytime MB improved from -5.62 to -2.42 from the nvPOA to the BASE

case, consistent with Woody et al. (2016), while the correlation improved from 0.67 to 0.79. The remaining underprediction is qualitatively consistent with Ensberg et al. (2014), who concluded that current estimates of SOA mass from gasoline and diesel emissions cannot produce the SOA observed in Pasadena if unaccounted for non-vehicular sources contribute heavily to the pool of SOA precursors. Positive matrix factorization (PMF) analysis of the observed OA into surrogate components revealed the strong influence of semivolatile oxygenated organic aerosol (SV-OOA, dominated by compounds emitted locally)

in generating the afternoon OA loadings Hayes et al. (2015). The pcSOA approach reproduced this buildup of daytime urban SOA much closer than the nvPOA case.

The BASE case performed similarly to the nvPOA case in Pasadena at night. There, the PMF analysis indicated that the low-volatility oxygenated organic aerosol (LV-OOA) surrogate dominated with roughly constant diurnal-averaged concentrations (~2 µg m$^{-3}$) (Hayes et al., 2015). LV-OOA had a significant contribution from regional (non-urban) biogenic OA. Other

OA components like HOA, cooking influenced (CIOA) and locally sourced (LOA) contributed less throughout the day (~0.5-1 µg m$^{-3}$) although the concentrations of CIOA were elevated at night. The lack of significant observed OA growth at night/early morning and the lack of observed correlation between aerosol liquid water content and the fraction of condensed water soluble organic compounds indicated a less important role of nighttime chemistry due to nitrate radicals or aqueous phase uptake (Hayes et al., 2013; Zhang et al., 2012).

Aggregate daytime predictions at the Bakersfield site improved dramatically as well (Fig. 1). Bulk factor analysis of AMS and Fourier transform infrared spectroscopy (FTIR) observations indicated five principal $PM_1$ source categories including aromatic SOA (24%), alkane SOA (41%), OA formed at night (10%), SOA from petroleum operations (14%) and vegetative detritus (10%). Multiple studies have characterized the importance of organic nitrates for nighttime SOA composition at the



site (Liu et al., 2012b; Rollins et al., 2012; O'Brien et al., 2013), while daytime OA formation appears to be driven by oxidation of primary vapors followed by condensation (Zhao et al., 2013). Baker et al. (2015) showed that CMAQv5.0.2 with nonvolatile POA underpredicted $PM_{2.5}$ organic carbon concentrations at both Pasadena and Bakersfield, with fractional biases (FB) of -53% and -144% , respectively, consistent with these results from CMAQv5.2 with nonvolatile POA.

The BASE case underpredicted concentrations at Bakersfield at night, and this was partially due to missing particle-phase organic nitrate compounds (Rollins et al., 2012) that are not formed in the gas-phase by the CB05e51 mechanism used in the CAL simulation. Amine compounds contributed about 10% or less of OA at Bakersfield (Liu et al., 2012b) and Pasadena (Hayes et al., 2013); these are missing from the particle-phase representation in this version of CMAQ as well. The day/night trends are consistent with the results of WRF-Chem application to the same campaign Fast et al. (2014). Differences that

emerge include a stronger OA daytime peak predicted by CMAQ at Pasadena and higher nighttime concentrations predicted by WRF-Chem at Bakersfield. These differences may have resulted from the use of slightly different anthropogenic emission inventories and selection of different modules for PBL dynamics.

  At Sacramento, the nvPOA case predicted significantly higher total OA concentrations at night (MB = 0.48 μg m$^{-3}$) than during the day (MB = -1.13 μg m$^{-3}$), while the BASE model agreed better with observations (nighttime MB = -0.35; daytime

= -0.41 μg m$^{-3}$). The magnitude of the day/night performance for the rural Cool site was encouraging (MB = -0.46 and -0.05, respectively), as was the correlation (r = 0.74 and 0.63, respectively). Setyan et al. (2012) isolated three factors using PMF analysis at Cool including hydrocarbon-like organic aerosol (HOA) (9% of total OA), a less-oxidized oxygenated OA (LO-OOA, 50.3%) that peaked in the early evening and a more-oxidized oxygenated OA (MO-OOA, 40.7%) that stayed relatively constant throughout the day with slightly elevated concentrations at night. The authors concluded that the LO-OOA

was correlated with urban transport while the MO-OOA was likely influenced by biogenic VOC oxidation. The fact that about 60% of the total OA mass was related to urban sources helps explain why the introduction of POA partitioning and pcSOA mass in CMAQ would make a difference to predictions at the Cool site, in the absence of regional fire events. As urban plumes move downwind, pcSOA and POA aging in the BASE case replace most of the evaporated POA and may add extra mass to the total OA burden. For example, the additional anthropogenic OA mass improved model performance at the Cool site, which is

often directly downwind of Sacramento, but had less impact on the rural Centreville, AL site.

  The BASE case, run for the EUS domain (Fig. 2), significantly underpredicted observations at both Centreville, Look Rock and Yorkville (daytime MB = -2.42, -3.39, and -0.66 μg m$^{-3}$ respectively) although the OA update does improve predictions slightly compared to the nvPOA case (daytime MB = -2.72, -3.78 and -1.16 μg m$^{-3}$ respectively). Correlation coefficients improved at all locations during both day and night, with one exception at Centreville during night. Biogenic SOA was the

main OA component during SOAS (Xu et al., 2015), and CMAQv5.2 potentially under-represents these compounds due to uncertainties in NO$_x$-dependence, oxidant loadings, and missing organic-water interactions (Pye et al., 2017), as they are not yet included in this version of the model. Although previous studies have demonstrated vapor wall-losses to be unimportant for SOA formation from $\alpha$-pinene ozonolysis (Nah et al., 2016) and $\beta$-penene oxidation by nitrate radicals (Boyd et al., 2015), it is unclear to what extent vapor losses may influence SOA yields from other monoterpene and sesquiterpene precursors or

from reaction with OH. Further studies on vapor wall-losses are necessary. The BASE case predicts that the composition of





total OA at Centreville was 65% biogenic SOA, 30% combustion SOA, and 5% POA. A prior simulation using GEOS-Chem and a similar implementation of the SIMPLE parameterization for urban and biomass burning SOA identified a contribution of 28% of the OA from urban sources, which was consistent with the fossil fraction of the carbon at the Centreville site (Kim et al., 2015). Predictions from the BASE model are similar, especially since combustion SOA and POA are from both anthropogenic and biomass burning combustion sources. The model predictions for the EUS domain include semi-explicit treatments of isoprene SOA (Pye et al., 2013) and terpene nitrate SOA formation (Pye et al., 2015), following the approach of (Pye et al., 2017).

The inclusion of semivolatile POA and pcSOA did not lead to much change in southeastern U.S. urban centers, Birmingham and Atlanta, but there was improvement at night in both locations (MB reductions of ~50%). Pye et al. (2017) estimated that primary organic carbon was overpredicted by a factor of 1.8 in the southeast U.S., thereby compensating for the underestimated SOA in that study.

Figure 3 compares CMAQ OA species to AMS factors derived from PMF analysis. For the nvPOA case, POA was calculated as the sum of CMAQ species POC and PNCOM and evaluated against observed HOA, even though some fraction of the material is aged. Simon and Bhave (2012) showed that this aging is predicted to add a modest amount of non-carbon mass to POA in California (~20% or less). For the BASE case, POA was summed from the fresh primary organic emissions (LVPO1, SVPO1, SVPO2, SVPO3, and IVPO1) and evaluated against HOA. This model-observation comparison clearly indicates, as many past studies have, that most OA, even in urban areas is partially or heavily oxygenated and that models using an approach like the nvPOA case replace OA formed by secondary processing with primary emissions. Thus, those models get an approximately correct estimate of OA concentrations for the wrong reasons and often with the wrong diurnal profile (De Gouw and Jimenez, 2009) (see section 4.2). At Pasadena, the nvPOA case overestimated HOA concentrations by a factor of 2.0. Treating the POA as semivolatile in the model resolved this discrepancy by reducing the predicted HOA factor. The combination of primary vapor aging and introduction of pcSOA resolved the OOA underprediction (MB improved from -4.45 to -0.81 µg m$^{-3}$. Figure 3 shows similar results at Bakersfield and Cool: average HOA predictions were reduced dramatically from the nvPOA to BASE case while OOA predictions increased but still underpredicted observations by -1.3 and -0.28 µg m$^{-3}$. The underprediction disappears if the HOA and OOA observed and predicted concentrations are normalized by the CO enhancement ($\Delta$CO; see supporting information, Fig. S2). Because normalizing by $\Delta$CO reduces the influence of errors in transport and dilution, this metric better isolates the performance of the specific chemical and microphysical processes under investigation here. However, confidence in CO model performance from CMAQ is uncertain and should be evaluated regularly and in greater detail in the future. Baker et al. (2015) and Woody et al. (2016) noted that the CO concentrations predicted by CMAQ during CalNex were systematically 10-30% low, and potentially emblematic of a general limitation of CMAQ capabilities for CO prediction.

Both Hayes et al. (2013) and Woody et al. (2016) used the slope of OOA as a function of odd oxygen ($O_x = O_3 + NO_2$) as an indicator of how well the model captured the general magnitude for urban SOA formation and its dependence on available oxidants. Hayes et al. (2013) found this slope at Pasadena to be 146 µg m$^{-3}$ ppbV$^{-1}$, which is more consistent with the BASE model prediction (103 µg m$^{-3}$ ppbV$^{-1}$) than with the nvPOA model (7 µg m$^{-3}$ ppbV$^{-1}$). While the BASE model does better in capturing urban OOA, it still underpredicts its magnitude of OOA formation by 35% according to this metric.





The underprediction appears more pronounced at the highest $O_x$ levels, according to Fig. 4. Woody et al. (2016) found similar improvement in this metric when applying a version of CMAQ that explicitly included aging of first-generation anthropogenic OOA (slope equal to $72 \mu g\ m^{-3} ppbV^{-1}$). They noted that CMAQ predictions could match the observed slope well if reactive anthropogenic SOA precursor vapors were added to account for missing S/IVOCs. It is not surprising that the BASE model per-

formance falls between these two cases since the pcSOA model species was designed to account for these and other uncertain SOA formation processes. Model performance between the nvPOA and BASE cases improves similarly at Bakersfield ($m_{obs}$ = 80 μg m$-3$ppbV$^{-1}$; $m_{svPOA}$ = 54 μg m$-3$ppbV$^{-1}$) and Cool ($m_{obs}$ = 75 μg m$-3$ppbV$^{-1}$; $m_{svPOA}$ = 64 μg m$-3$ppbV$^{-1}$). As with the HOA/OOA comparison, normalizing by $\Delta CO$ improves the model performance even further, possibly canceling errors in meteorological phenomena and dispersion processes (see supporting information, Fig. S3). Because CO performance

is uncertain in CMAQ as previously stated, a detailed examination of the sensitivity to the pcVOC emission scale factor and OH reaction rate constant is useful.

### 4.2    Model sensitivity to uncertain pcSOA parameters

Figure 5 shows diurnal-averaged profiles for the observations and several model cases including nvPOA, BASE, HEBR and LEBR. The cases with pcSOA captured the variability throughout the day, including the timing of the afternoon OA peak, and the day-to-day variability, at all California sites better than the nvPOA case. At Pasadena, The mean bias improved from

-4.01 to -2.48 μg m$^{-3}$ and the correlation coefficient increased from 0.15 to 0.81. Performance at Sacramento was particularly improved in the BASE case (MNE = 0.24; r = 0.91). The pcVOC emission scale factors for HEBR and LEBR were set to the upper and lower bounds of the optimal parameter set reported by Hayes et al. (2015). The HEBR case predicted the magnitude of the afternoon OA peak at Bakersfield, Sacramento, and Cool, but still underpredicted the peak at Pasadena. However, that

underprediction is consistent with the underprediction at night, and so may be a result of missing nighttime sources or processes. The effect of perturbing pcVOC emissions on concentrations at Centreville and Look Rock was more subdued. At those sites, pcSOA contributed 24% and 36% to the total OA mass in the $PM_1$ range, respectively, in the BASE case. Generally, pcSOA concentrations were reduced by 30% when the LEBR parameters were employed. Accounting for the pcSOA reduction and the effect of decreased total OA absorbing media on other OA surrogates resulted in a 13% decrease in OA $PM_1$ concentrations.

We then varied the both pcSOA parameters in the BASE model configuration as described in table 4 to quantify the sensitivity of model predictions holistically. Figure 6 shows the NMBF and NMEF metrics calculated for each combination of input parameters investigated as well as those of the nvPOA run. These factors were computed for model-observation pairs grouped by each station individually, and then averaged together. The conclusions of this analysis did not change when these metrics are calculated for all the pairs without grouping into sites first. When comparing against continuous data collected during CalNex

and CARES, the nvPOA case performed worse than every sensitivity case for both metrics. For NMBF, the optimal emission factor likely lies between the base and high values, as long as the appropriate OH reaction rate is chosen to constrain the total mass formed. It is not clear whether the parameter-space explored constrained the NMEF values although they may possibly plateau at 0.48 as one moves up and to the right on Fig. 6. These findings are consistent with those of Hayes et al. (2015), although that analysis parameterized the VOC precursor emissions to the emission of CO rather than POA. The emission factor





used here avoids the potentially problematic CO model predictions by relying on the more robust POA emission rate as an indicator. An area of future work will be to develop the capability to use total VOC emissions from all combustion sources as the basis for pcVOC emissions since they should be even more correlated.

Evaluation of NMBF and NMEF for the CONUS11 simulations (Fig. 7) result in different optimal pcSOA parameters. Daily-averaged model predictions are paired with observations at sites from both the IMPROVE and CSN networks for January and July 2011. For this application, the nvPOA case results in lower NMEF (0.83) than half of the sensitivity cases, which are those with emission factors or reaction rates higher than those of the BASE model (NMEF = 0.82). Further, the nvPOA case has a lower NMBF (0.39) than all cases but the LEBR case (0.27). The parameter-space is unbounded here and even lower emission factor/reaction rate combinations could potentially reduce the NMBF and NMEF, but those solutions are not supported by the analysis for the CAL domain. The need for higher emissions or reaction rates to support higher OA concentrations at the California sites is not corroborated by the CONUS11 data, indicating that differences in the sources or production pathways that drive OA concentrations at these two scales is not completely captured with the uniform application of pcSOA. In the eastern U.S. for example, relative humidity or acidity may play a larger role for OA formation and deposition. Meanwhile in the wintertime cases, large wood burning area emission sources may not emit SOA precursors consistent with pcSOA formation. If these sources are significantly overpredicted, then lower pcSOA production rates yield better agreement at the CONUS scale for the wrong reasons. This possibility will be explored by applying a source-resolved emission inventory determine how urban SOA formation from vehicles and cooking sources might be better simulated in the future.

## 4.3 Evaluation against routine-monitoring data

We extended the analysis from specific measurement campaigns to a thorough evaluation of a year-long dataset at the continental U.S. scale, CONUS11. For this application, we used daily-averaged results from the low-emission base-reaction (LEBR) model case, which performed best in the January and July CONUS11 scenarios (discussed in section 4.2). Figure 8 summarizes the model performance across both the IMPROVE and CSN networks for the entire year. The MB across all CSN model-observation pairs decreased substantially from the nvPOA (0.88) to the LEBR (0.39) case, while the MB at IMPROVE sites stayed roughly the same. The aggregate improvement across all sites resulted in about a 50% reduction in MB, a 12% reduction in mean error, and very little change to the correlation coefficient. As shown in Fig. 8c, error decreases occurred at a majority of sites, with large error decreases at some sites (up to and exceeding $3 \, \mu g C \, m^{-3}$). A minority of sites showed small increases in error, but none showed increases above $0.5 \, \mu g C \, m^{-3}$. Segregating the data regionally (Fig. 8d) indicated similar changes to the model OC distribution across the U.S. The new model yielded slightly less variable OC values (i.e. shorter whiskers) and only small changes to the median OC predictions. Changes to the explanatory power of the model data were mixed with some regions indicating better correlations (Northeast, Southeast, Plains), some worse correlations (Midwest, Southwest) and some staying the same (Northwest). The consistency between the performance of the nvPOA and LEBR cases for the CONUS11 domain was likely due to the coordination of several confounding factors including spatio-temporal variability, uncertainty in emissions and reactivity, and meteorological errors (e.g. wet deposition, boundary-layer dynamics). The evaluations presented in section 4.1 demonstrated the extensive influence of temporal averaging: at many sites throughout the





CONUS domain, the nvPOA case likely overestimated OC concentrations at night and underestimated them during the day, and these compensating errors averaged out. This stresses the critical need to evaluate models like CMAQ at multiple spatial and temporal scales where data are available.

Figure 9 illustrates the role of seasonal variability. The nvPOA case overpredicted daily-averaged OC concentrations in the winter months and underpredicted them in the summer months with correlation coefficients in the range of 0.3-0.64. This bias trend is a well-documented feature of models that employ the nvPOA in general (Shrivastava et al., 2008). The LEBR case reduced almost all of the bias in October-December 2011 and some of it in January and February, possibly a result of the more accurate temperature-dependence in the semivolatile POA model. POA plays a larger role in the winter months than in the summer because of favorable temperature-dependent partitioning. However, without the pcSOA species, the LEBR case would likely have underpredicted slightly the observations in the winter time. Meanwhile, performance during the summer months improved slightly for the LEBR case, with significant reduction in bias for July. The transition season months showed smaller bias reductions, with 0.22, 0.14, 0.12, and 0.14 $\mu g\ m^{-3}$ bias reductions for March, April, September and October, respectively. During these months, the opposing changes induced by POA volatilization and pcSOA formation had highly variable impacts across the U.S., with dependence on temperature, boundary-layer height, and oxidant loadings. The correlation coefficient did not change appreciably from the nvPOA case to the LEBR case, in contrast to the dramatic changes in correlation seen when comparing against California hourly data. A regional evaluation of the NMBF and NMEF metrics broken down by season illustrates that performance in almost every region of the U.S. improved throughout the year from the nvPOA to the LEBR case (Fig. 10). The nvPOA case had the worst-performing predictions during summer and the LEBR updates moved all but the northeast region into the statistical area defined as acceptable by Yu et al. (2006). Performance during the summer improved for all regions and improved substantially for the northwest. The NMBF was reduced for all regions in the fall scenarios although the NMEF changed only slightly. The only degraded performance occurred in the northwest in spring, where the NMBF increased by about a factor of 2 and the NMEF by a factor of 3. This is likely driven by errors in the emissions and aging of POA from fire events, and should be investigated in the future as more is learned about the identity and fate of the compounds emitted by this group of sources.

## 4.4 OA spatio-temporal composition in CMAQ

The total OA surface concentrations predicted by the model vary considerably from winter to summer for many regions of the nation (top row of Fig. 11). In the winter, urban locations emerge as the sources mainly driving the regional OA distribution, whereas in the summer, these urban centers compete with heavily forested regions of the country like the southeast and the Sierra Nevadas. Although the abundance of biogenic OA observed for six weeks at the Centreville site (Xu et al., 2015) is qualitatively reproduced by the model, significant uncertainties exist in our knowledge of the air-surface exchange of important biogenic precursors, their SOA yields, and the partitioning properties of their oxidation products. The annually-averaged contribution of POA (unreacted primary compounds in the particulate phase) predicted by the model is in the range of 5-10% increasing to 30% in some cities, especially at high elevations with colder temperatures, which is in the same range as observations (Jimenez et al., 2009). These maxima are driven by wintertime episodes when the atmospheric lifetime of POA





against evaporation and gas-phase oxidation increases substantially. The LEBR model predicted that in the summer there is little contribution of these fresh emissions to particulate loadings as most of the mass was predicted to be oxidized quickly in the gas phase. CMAQv5.2 with the LEBR configuration should be evaluated against AMS-inferred HOA concentrations at cities characterized by higher wintertime POA fractions, (e.g. Xu et al. (2015)).

pcSOA concentrations are driven both by variability in primary pcVOC emissions and by efficient photochemical oxidation, as illustrated by Fig. 11 (third row). The summertime concentration field was similar to that of the total OA concentrations and was relatively dispersed in the eastern U.S. like other regional secondary pollutants (e.g. ozone, particulate nitrate). Interestingly, the wintertime concentration field in the east was more dispersed than in the summer because of the longer lifetime of pcVOC against oxidation to pcSOG. The increased role of residential wood combustion emissions led to elevated pcSOA
concentrations in the northeast U.S. compared to those in the summer. The pcVOC emission sources in the western U.S. were visually evident from the pcSOA concentration field in both the summer and winter simulations, mostly due to relatively less population density in western states compared to the east. An interesting dynamic associated with pcSOA appeared in California. In the summer, enhanced photochemistry caused high pcSOA concentrations in southern California ( 4 µgC m$^{-3}$). Meanwhile concentrations were elevated but not as much in the San Joaquin Valley. The relationship was flipped in the winter-
time where concentrations were low in southern California and much higher in the valley. The wintertime San Joaquin Valley enhancement was likely due to lower boundary layer heights enhancing the oxidative capacity of the airshed and increasing the formation rate of pcSOA. Future work will investigate episodes like this to determine if more can be learned about how the contributions of various SOA pathways vary seasonally. The percent change in POA concentrations from the nvPOA to the LEBR cases was highest during the summer and far from urban sources, as expected. For this study, the POA boundary
conditions were renamed SOA and highly affect the percent change in POA close to the model boundaries. In the summer time, the change in POA concentrations in the domain interior varied from -100% to -70% change in California. The winter simulations showed more variability (-100% to -55% change), indicating that even when treating POA as semivolatile, it is still an important component of urban and suburban particulate loadings, and important to consider in exposure assessments.

### 4.5 Multiyear organic carbon trends

For many policy and environmental applications, it is important for the model to capture trends in ambient concentrations in response to emissions changes. In order to probe CMAQv5.2 sensitivity to changes in real-world emissions from 2002 to 2011, we applied the model (both the nvPOA and LEBR cases) to the continental U.S. domain during January and July 2002 (CONUS02) and compared the modeled OC predictions to available CSN and IMPROVE observations (Fig. 12). The OC trend for IMPROVE sites between January 2002 and January 2011 were relatively flat and the model reproduced the behavior for both
cases. The January CSN site trends were more complex. While the observations indicated a decrease of 0.92 µgC m$^{-3}$ yr$^{-1}$, both the nvPOA and LEBR model cases predicted increases of 0.87 and 0.65 µgC m$^{-3}$ yr$^{-1}$, respectively. Because this deviation occurs in the context of wintertime scenarios, uncertainties from emissions or misrepresented boundary layer dynamics likely play a more important role. With more realistic emissions from this source, both CMAQv5.2 model configurations would predict greater OC, potentially overpredicting the observations by a similar magnitude as the 2011 datasets.





Both CSN and IMPROVE sites during July show a similar trend, although it is exacerbated for CSN sites. Here, the observations indicate dramatic OC decreases (-5.72% $yr^{-1}$) while the nvPOA and LEBR models predict increases by 4.28% and 3.65% $yr^{-1}$, respectively. Clearly the model is missing an important source or pathway for OC formation in the 2002 simulation. The 2011 predictions overlap reasonably with the observations, so this pathway either diminishes substantially when

moving to the 2011 scenario or there is a coincident offset by other pathways that may be overestimated by the model in 2011. The 2002 CSN observations are potentially biased by the application of the National Institute for Occupational Safety and Health (NIOSH) method; the IMPROVE (2002 and 2011) and the CSN 2011 observations were gathered using the IMPROVE method, which is consistent with the characterization of the POA emission factors used to build the emission inventory. Because the differences between the NIOSH and IMPROVE methods affects the observed ratio of OC to elemental carbon (EC),

the potential magnitude of OC bias can be estimated by inspecting the bias in EC. For the CSN 2002 dataset, the MB for EC is 0.2 $\mu g\,m^{-3}$, indicating that most of the underprediction at CSN sites in summer 2002 is not related to discrepancies between measurement methods.

One possible formation pathway of particle-phase OC mass that could have been higher in 2002 involves perturbations of aerosol liquid water concentrations. Carlton and Turpin (2013), Pye et al. (2013) and Pye et al. (2017) argued that higher aerosol

water concentrations, possibly resulting from increased sulfate concentrations in the past, could have enhanced the partitioning of semivolatile organic compounds depending on their solubility, or it could have enhanced reactive uptake rates of VOC oxidation products. This is consistent with Xu et al. (2016), who showed that 88% of OA were water-soluble at the Centreville site during SOAS and that only 25-40% of the total water-soluble organic compounds were in the particle phase throughout the day. Figure 13 indicates the trend toward higher OC biases as sulfate increased. This trend occurred despite the fact that sulfate

and elemental carbon aerosol predictions did not underestimate observations as did OC predictions. CMAQv5.2 includes the aerosol-water dependent pathway of isoprene OA formation via IEPOX processing. A similar interaction among products of anthropogenic VOC oxidation may be responsible for the discrepancy in the urban-influenced CSN sites. Another potential OC pathway, organic nitrate formation, could have played a larger role in 2002 and diminished over the last decade due to decreased $NO_x$ emissions (e.g. Pye et al. (2015)). Those authors showed that a 25% reduction in $NO_x$ led to a 25% reduction

in organic nitrate SOA. Since 2002, $NO_x$ concentrations in the U.S. have decreased due to the implementation of the $NO_x$ Budget Trading Program (or $NO_x$ SIP Call and the Clean Air Interstate Rule. For the combined 2002/2011 CSN dataset, the OC underprediction improved from almost -5 $\mu gC\,m^{-3}$ at 50-60 pbbV $NO_x$, to unbiased at 10 ppbV $NO_x$ and less . Although these relationships are not mechanistic proof of a relationship between OC aerosol formation and $SO_2$ or $NO_x$ emissions, they do encourage further scrutiny of potential co-benefits that reducing the latter two pollutants has on OA burden and thus

public health. Finally, it is possible that the ratio of POA to intermediate and high volatility unspeciated organic combustion emissions changed from 2002 to 2011 as new emissions controls and fuel formulations have been adopted (May et al., 2014). Thus the analysis presented here may speak to the robustness of using the pcSOA approach for regulatory applications. Fully quantifying the impacts of the pcSOA assumption would require comparing to simulations include both speciated, source- and year-specific NMOG estimates in the emissions input generation process and accurate speciated SOA yields in the CTM.



## 5 Conclusions

We have shown that CMAQv5.2 with a semivolatile representation of POA compounds and an introduction of the lumped pcSOA model species predicts organic aerosol concentrations acceptably against high-time-resolution field campaign data and daily-averaged routine-monitoring network products. In almost all cases, the model improves diurnal and seasonal trends compared to the model with nonvolatile POA, a consequence of better representation of both temperature sensitivity and photochemical oxidation dependence. Because the BASE model is able to achieve similar performance to other more-complicated approaches to OA formation, we advise the OA fate and transport modeling community to consider adding complexity only when there is enough experimental data to provide an associated reduction in uncertainty.

The discrepancy between the optimal parameters inferred from the CAL and CONUS11 simulations highlights the importance of not overstating the meaningfulness of pcSOA or the value of the optimal parameters presented here. Because these parameters are fit to available observations, they lose some explanatory power as one extrapolates to other domains or time periods. However, the essential features of modeled pcSOA formation, the oxidation of a VOC precursor followed by condensation to the particle phase, broadly represent the transformations of most of the likely pathways to OA formation needed by the model. This approach parallels the more detailed, but still very uncertain, approaches used in similar CTMs simulating OA from primary sources (Fast et al., 2014; Koo et al., 2014) while maintaining consistency with existing regulatory inventories developed over decades. For these reasons, we feel confident that CMAQv5.2 with semivolatile POA and pcSOA is a substantial step toward a more accurate representation of OA formation, even though future work will focus on replacing the pcSOA species with specific mechanisms constrained by direct measurements.

Based on the limitations discussed in this work, we recommend the chemical transport modeling community address the following concerns:

- The entire volatility spectrum of primary organic emissions, including IVOCs, should be incorporated directly into emission inventories and applied to model input as a function of relevant parameters like combustion technology, fuel type, etc.

- Because of their unique properties and variable OA formation potential, organic compound emissions from biomass burning sources will likely need to be treated distinctly from fossil fuel combustion sources in CTMs. For wildfires the net added mass due to SOA formation, relative to the initial POA mass, is 1-2 orders-of-magnitude lower than for urban sources (Cubison et al., 2011).

- Aerosol-water interactions are an important complexity to incorporate consistently in CTMs. These interactions not only impact OA loading Pye et al. (2017) but have important implications for deposition rates, diurnal profiles, multiyear trends, and model response to simulated control strategy.

- Organic nitrates were detected at multiple sites during the CalNex, CARES and SOAS campaigns. A consistent treatment of these compounds from anthropogenic and biogenic carbon sources and the effect nitrate-group addition has on volatility and solubility is needed.




- In general, greater efforts are needed to improve the conceptual link between the gas- and aerosol-phase chemical components in CTMs. An example of this approach is described for isoprene epoxide degradation by Pye et al. (2013) and monoterpene nitrate formation by Pye et al. (2015).

We further identify the following potential measurement efforts as useful for addressing critical gaps in our current understanding of OA formation:

- Winter-time ambient observations of urban OA in multiple cities with varying meteorology. Observations are needed at high time-resolution (at or greater than 1 hour) in order to discern important sensitivities to photooxidation, temperature, and atmospheric water content. Important properties to constrain are POA fraction, volatility/solubility, and aging time-scales.

- Detailed transport model comparisons to existing and upcoming measurements of ambient biomass burning plumes including characterization of potential downwind SOA enhancement as well as aging effects on volatility/solubility (e.g. Forrister et al. (2015)).

- Updated quantitative estimates of SOA yields as a function of volatility/solubility that take into account losses of low-volatility vapors to chamber walls.

We conclude that the important OA mass formation processes are most likely regional in scale, and that OA concentrations tend to be regionally distributed in the U.S., especially in summer. However, we also emphasize that despite the diminished contribution direct POA emissions have in the model predictions as a result of evaporation, they are still a critical component to consider, especially for exposure assessments near primary sources, urban centers, and in the winter. Moreover, the evaporated POA compounds likely react in the vapor phase and may contribute to particle concentrations again quickly or they may contribute to the formation of oligomer species, about which little is known with certainty.

The multiyear trends in Figs. 12 and 13 highlight the difficulty in integrating OA formation and properties into models to be used for regulatory applications. If interactions with regulated pollutants (e.g. $SO_2$, $NO_x$) is an important contributor to OA formation, then the applicability of a model configuration to a specific scenario may change over time as the concentration of those pollutants decreases. Pathways that used to be important may become less so and vice-versa, thus demonstrating the benefit of evaluating the model for a wide array of atmospheric conditions. Characterizing model results with this dynamic view helps improve the robustness of the model over time and guide future policy decisions and research.

## 6   Code and Data Availability

The CMAQv5.2 code for both the semivolatile and nonvolatile POA options is available via GitHub repository (https://github.com/usepa/cmaq). Data included in figures is accessible through the EPA ScienceHub portal (https://catalog.data.gov/dataset/epa-sciencehub). CalNex measurement data is available through the Earth System Research Laboratory data portal (https://www.esrl.noaa.gov/csd/groups/csd7/measurements/2010calnex/); CARES data through the DOE Atmospheric Radiation Measurement program portal (http://www.archive.arm.gov/discovery/#v/results/s/fiop::aaf2009carbonaerosol) and SOAS data through the Southeast Atmosphere Study website (https://esrl.noaa.gov/csd/groups/csd7/measurements/2013senex/).



*Acknowledgements.* The U.S. EPA, through its Office of Research and Development, supported the research described here. It has been subjected to Agency administrative review and approved for publication, but may not necessarily reflect official Agency policy. The authors gratefully acknowledge Drs. Heather Simon (Office of Air Quality Planning and Standards) and Brian Eder (National Exposure Research Laboratory) from the U.S. EPA for their helpful comments during the Agency review process. JLJ was supported by EPA STAR 83587701-0 and NSF AGS-1360834. SL and LR were supported by California Air Resources Board (contract 09–328). Funding for data collection during the CARES campaign was provided by the Atmospheric Radiation Measurement (ARM) Program sponsored by the US Department of Energy (DOE), Office of Biological and Environmental Research (OBER). RAZ was supported by U.S. DOE's Atmospheric Systems Research (ASR) program under Contract DE-AC06-76RLO 1830 at Pacific Northwest National Laboratory. QZ and AS were supported by DOE ASR DE-FG02-11ER65293. LX and NLN acknowledge National Science Foundation (NSF) grant 1242258 and US Environmental Protection Agency STAR grant RD-83540301.



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



**Table 1.** Properties of semivolatile POA species in CMAQv5.2.

| Species Name (Particle/Vapor) | Mol. Weight g mol$^{-1}$ | C$^{*,1}$ µg m$^{-3}$ | N$_C$ | O:C | H$_{vap}$ kJ mol$^{-1}$ |
|---|---|---|---|---|---|
| ALVPO1/VLVPO1[2] | 218 | 0.1 | 13.0 | 0.185 | 89 |
| ASVPO1/VSVPO1[3] | 230 | 1 | 14.5 | 0.123 | 85 |
| ASVPO2/VSVPO2 | 241 | 10 | 16.0 | 0.073 | 81 |
| ASVPO3/VSVPO3 | 253 | 100 | 17.5 | 0.032 | 77 |
| AIVPO1/VIVPO1[4] | 266 | 1000 | 19.0 | 0.00 | 73 |
| ALVOO1/VLVOO1[5] | 136 | 0.01 | 5.0 | 0.886 | 93 |
| ALVOO2/VLVOO2 | 136 | 0.1 | 5.5 | 0.711 | 89 |
| ASVOO1/VSVOO1 | 135 | 1 | 6.0 | 0.567 | 85 |
| ASVOO2/VSVOO2 | 135 | 10 | 6.5 | 0.447 | 81 |
| ASVOO3/VSVOO3 | 134 | 100 | 7.0 | 0.345 | 77 |

[1] C$^{*}$ values are defined at reference temperature 298 K.

[2] A = Aerosol; V = Vapor; LV = Low Volatility; PO = Primary Organic

[3] SV = Semivolatile

[4] IV = Intermediate Volatility

[5] OO = Oxidized Organic

**Table 2.** Properties of potential SOA from combustion emissions (pcSOA) species and its precursor, pcVOC for the base case (BASE) CMAQ simulation.

| Property | Value |
|---|---|
| Molecular weight | 170 g mol$^{-1}$ |
| Emissions Scale Factor (pcVOC/POA) | 0.0568 mol g$^{-1}$ |
| OH oxidation rate constant (k$_{OH}$) | 1.25 x 10$^{-11}$ cm$^{-3}$ molec$^{-1}$ s$^{-1}$ |
| Condensable product saturation concentration (C$^{*}$) | 10$^{-5}$ µg m$^{-3}$ |

**Table 3.** CMAQ model scenarios used for this study.

| Name | Spatial Domain | Duration | Gas-phase Chemical Mechanism | Reference |
|---|---|---|---|---|
| CAL | Southwestern U.S. | 3 May - 30 Jun 2010 | CB05e51 | (Baker et al., 2015; Woody et al., 2016) |
| EUS | Eastern U.S. | 1-30 Jun 2013 | SAPRC07tic | (Pye et al., 2015) |
| CONUS11 | Continental U.S. | 1 Jan - 31 Dec 2011 | CB05e51 | (Appel et al., 2016) |
| CONUS02 | Continental U.S. | 1-30 Jan; 1-31 July 2002 | CB05e51 | (Bash et al., 2013) |



**Table 4.** Details of base and sensitivity simulation exploring pcSOA formation parameters.

| Simulation | pcVOC Emission Scale Factor (mol g$^{-1}$) | k$_{OH}$ (cm$^{-3}$molec$^{-1}$s$^{-1}$) | Model domains |
|---|---|---|---|
| Nonvolatile POA (nvPOA) | NA | NA | CAL, EUS, CONUS11, CONUS02 |
| Base emission, base reaction (BASE) | 0.0568 | 1.25 x 10$^{-11}$ | CAL, EUS, CONUS11 (Jan, July) |
| Low emission, base reaction (LEBR) | 0.0387 | 1.25 x 10$^{-11}$ | CAL, EUS, CONUS11, CONUS02 |
| High emission, base reaction (HEBR) | 0.07 | 1.25 x 10$^{-11}$ | CAL, CONUS11 (Jan, July) |
| Base emission, low reaction (BELR) | 0.0568 | 1.0 x 10$^{-11}$ | CAL, CONUS11 (Jan, July) |
| Base emission, high reaction (BEHR) | 0.0568 | 2.0 x 10$^{-11}$ | CAL, CONUS11 (Jan, July) |
| Low emission, high reaction (LEHR) | 0.0387 | 2.0 x 10$^{-11}$ | CAL, CONUS11 (Jan, July) |
| High emission, low reaction (HELR) | 0.07 | 1.0 x 10$^{-11}$ | CAL, CONUS11 (Jan, July) |




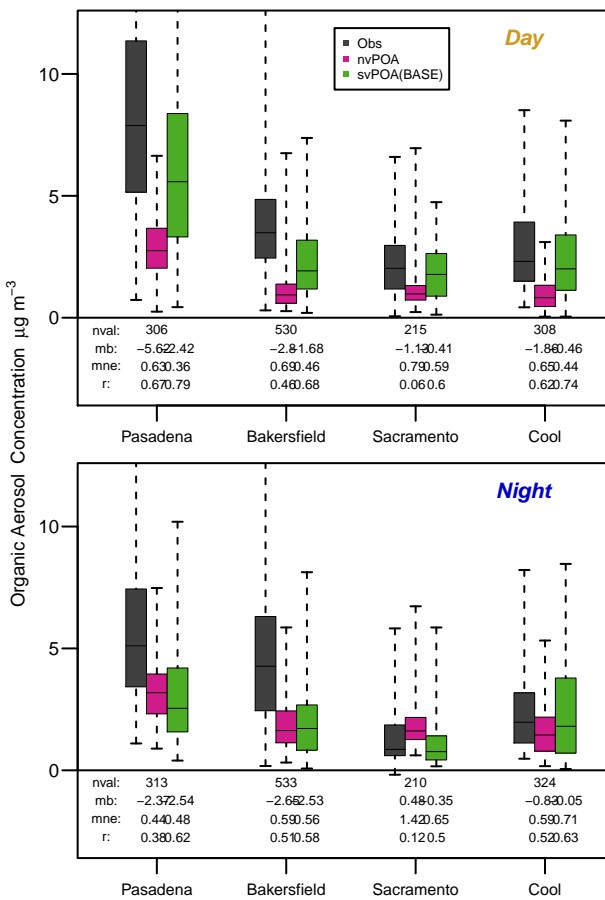

**Figure 1.** Organic aerosol concentrations (µg m$^{-3}$) observed (gray) with the HR-ToF-AMS at sites in California in 2010 (Pasadena, Bakersfield, Sacramento, and Cool). Also shown are the model predicted distributions at each site using the nonvolatile (pink) and base-case semivolatile (green) configurations. The top and bottom panels show data for daytime (0800-2000) and nighttime hours, respectively. Model values are projected to $PM_1$ to correspond roughly to the size cutoff of the AMS.




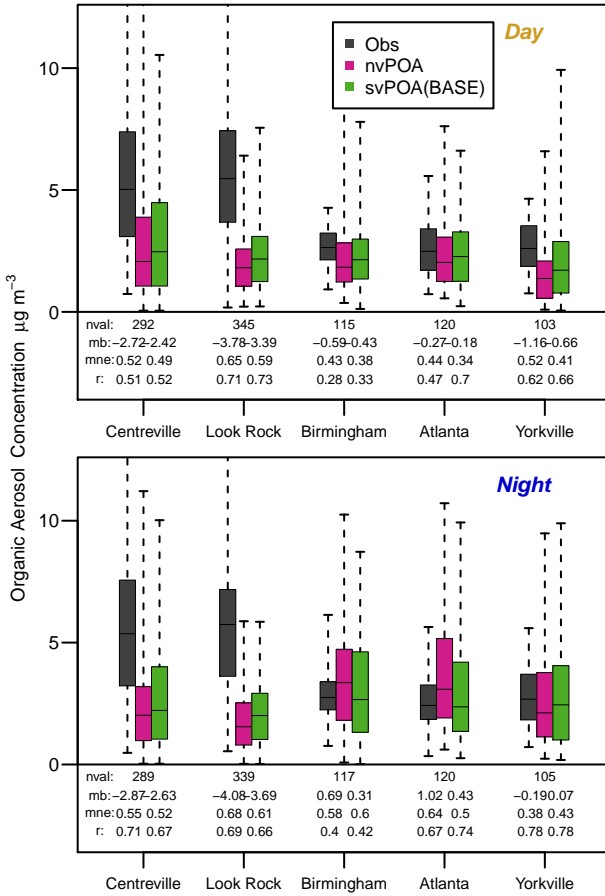

**Figure 2.** Observed (gray) and modeled (pink, green) organic aerosol (µg m$^{-3}$) and organic carbon (µgC m$^{-3}$) concentrations at sites in the southeast U.S. OA concentrations at the SOAS sites, Centreville and Look Rock, were measured with HR-ToF-AMS while OC concentrations at the SEARCH sites, Birmingham, Atlanta, and Yorkville were inferred as the difference between total carbon measured by ambient particulate carbon monitors and elemental carbon measured by aethalometers. Also shown are the model predicted distributions at each site using the nonvolatile (pink) and base-case semivolatile (green) configurations. The top and bottom panels show data for daytime (0800-2000) and nighttime hours, respectively. Model values for the SOAS sites are projected to $PM_1$ to correspond roughly to the size cutoff of the AMS, while for the SEARCH sites the sum of the Aitken and accumulation modes was applied. All model data is produced from the EUS simulation, which uses SAPRC07tic and occurs during June 2013.




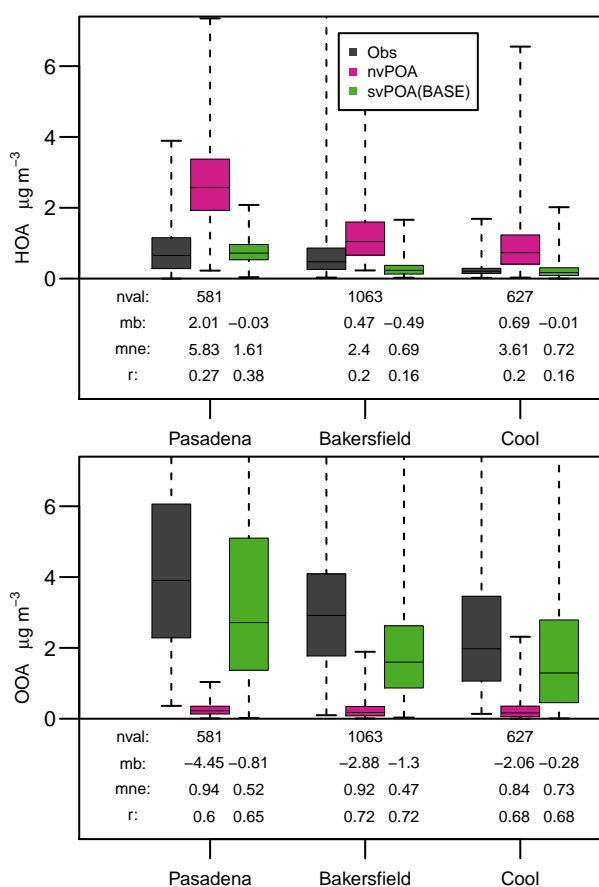

**Figure 3.** Comparison of OA factors derived from positive matrix factorization (PMF) of HR-ToF-AMS observations (gray) at sites in California (Pasadena, Bakersfield, and Cool). Also shown are the model-predicted concentrations of each factor at each site using the nonvolatile (pink) and base-case semivolatile (green) configurations. The top panel compares estimations for hydrocarbon-like OA (HOA) and the bottom panel for oxygenated OA (OOA).





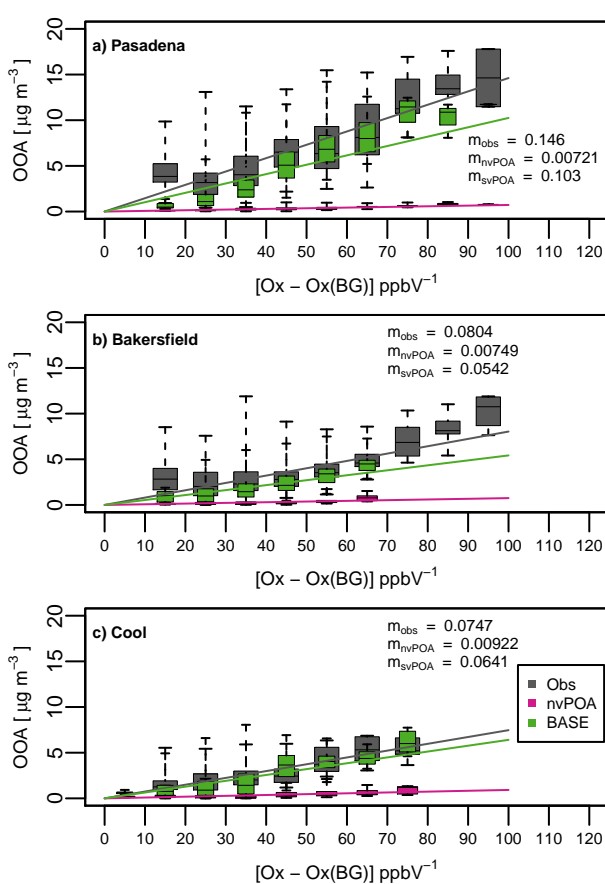

**Figure 4.** OOA concentrations ($\mu$g m$^{-3}$) at California sites as a function of the measured $O_x$ ($O_3$ + $NO_2$) concentration. A background concentration, $O_x$(BG), of 13.5 ppbV is assumed, consistent with Hayes et al. (2015). The boxes behind each trend indicate the 25th and 75th percentiles of the data. The whiskers extend to the 10th and 90th percentiles. The solid horizontal lines in each box identifies the median, and the solid curves indicate the means of each model run. Model values are projected to $PM_1$ to correspond roughly to the size cutoff of the AMS.



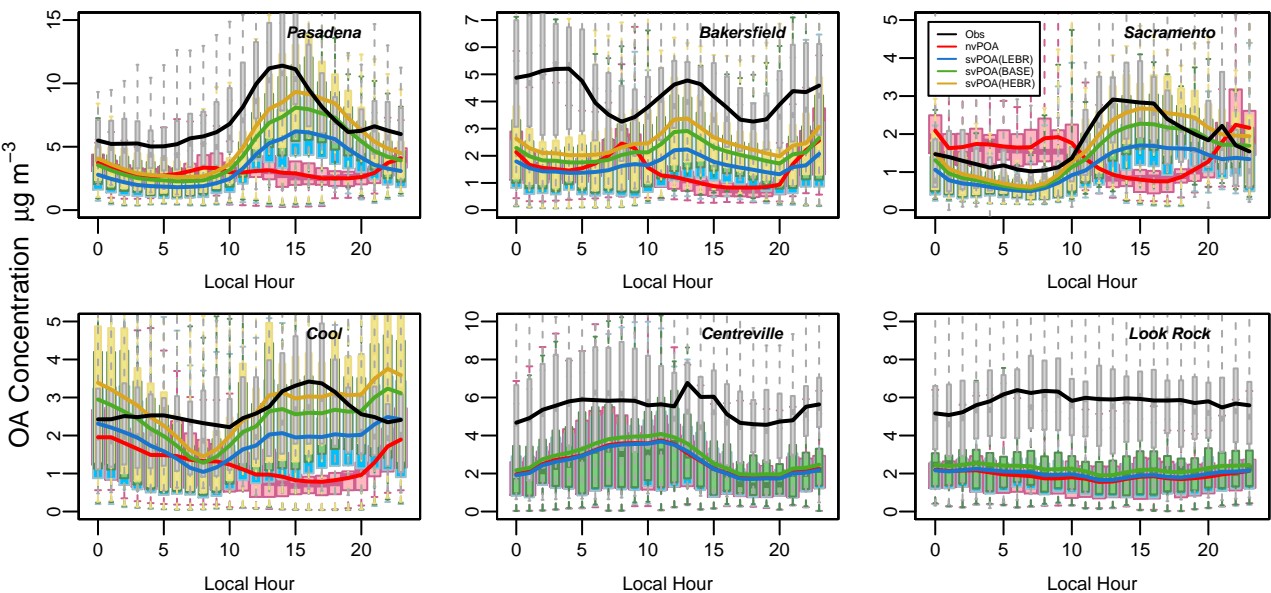

**Figure 5.** Diurnal OA concentration profile (μg m$^{-3}$) observed (black line, gray bars) with the HR-ToF-AMS at sites investigated during the CalNex, CARES, and SOAS campaigns. Also shown are the model-predicted distributions for the nvPOA case, BASE case and two sensitivity cases (LEBR and HEBR). The boxes behind each trend indicate the 25th and 75th percentiles of the data. The whiskers extend to the 10th and 90th percentiles. The solid horizontal lines in each box identifies the median, and the solid curves indicate the means of each model run. Model values are projected to $PM_1$ to correspond roughly to the size cutoff of the AMS.

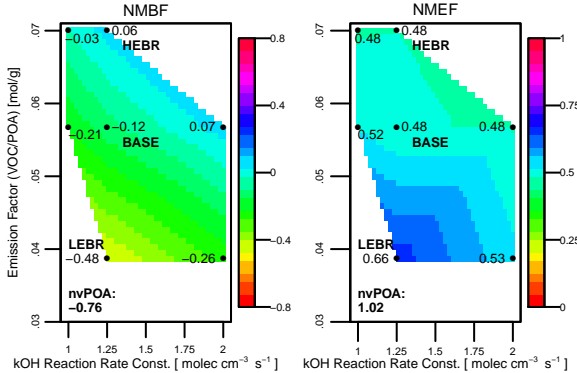

**Figure 6.** Evaluation of the effect of uncertain parameters for pcSOA formation, emission factor and OH reaction rate constant, for the CAL domain. The normalized mean bias factor (NMBF) and normalized mean error factor (NMEF) symmetrically represent both under- and overprediction.





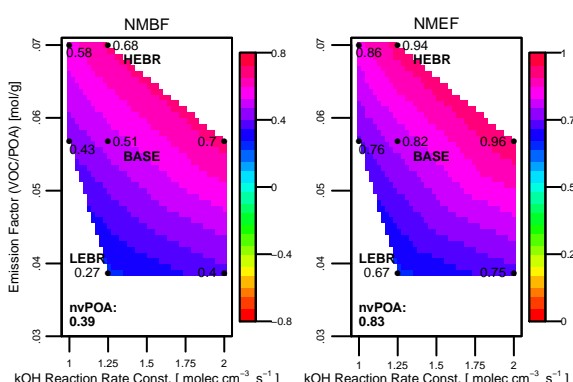

**Figure 7.** Evaluation of the effect of uncertain parameters for pcSOA formation, emission factor and OH reaction rate constant, against OC measurements from IMPROVE and CSN networks. Data from January and July only of the CONUS11 simulations were used for this analysis.




**Figure 8.** Continental-wide organic carbon (OC) evaluation of the nvPOA case and the best-performing sensitivity case (LEBR) against routine-monitoring data from IMRPOVE and CSN sites for an annual simulation during 2011. a) Distribution of OC observations from both networks individually and combined. b) Distributions of OC bias for each network and combined. c) Histogram of error changes from the nvPOA to the svPOA (LEBR) case at specific stations aggregated throughout the year. d) Regional distribution of OC observations and predictions throughout the annual simulation.



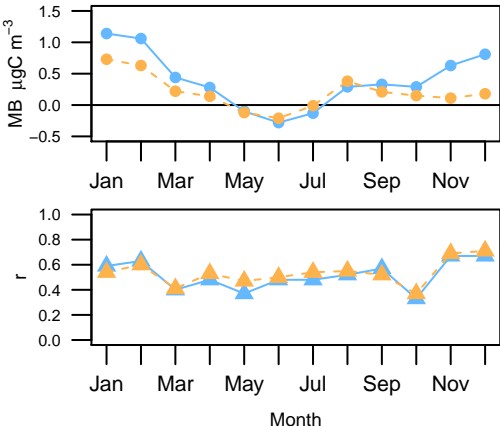

**Figure 9.** Monthly model performance for the nvPOA (blue) and LEBR (orange) cases throughout the 2011 simulation. The top panel indicates the mean OC bias while the bottom panel indicates the correlation coefficient of each model run with the CSN and IMPROVE observations.

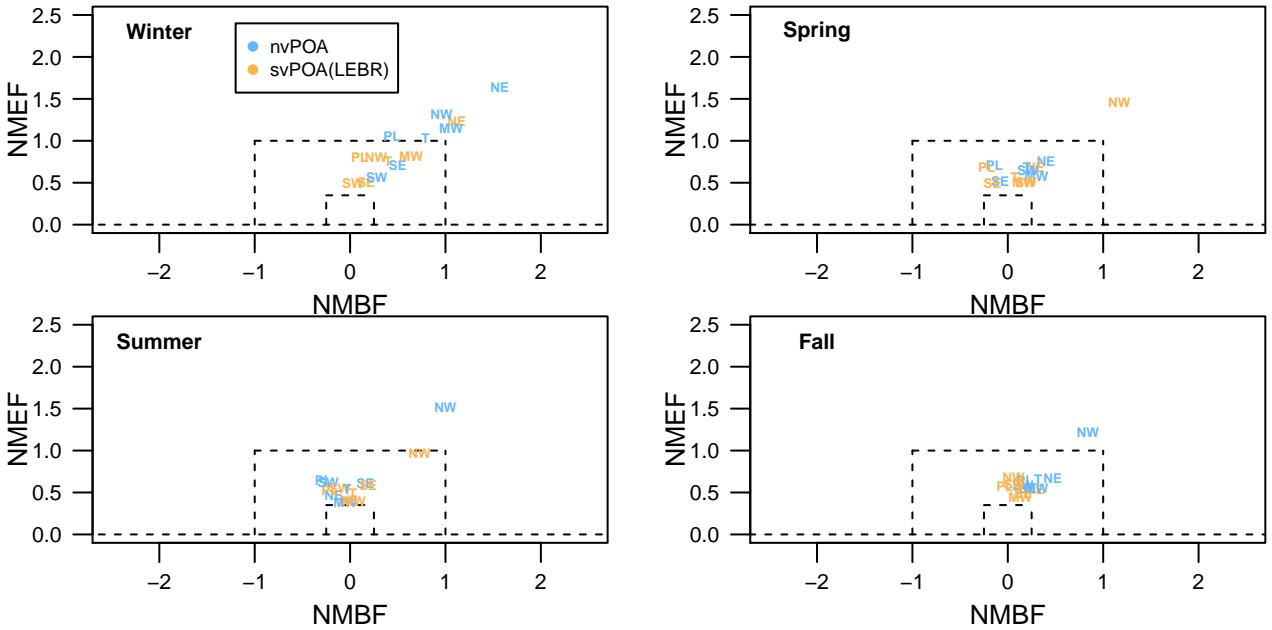

**Figure 10.** Regional and seasonal model performance at routine monitoring stations for the CONUS11 simulation as a function of normalized mean bias factor (NMBF) and normalized mean error factor (NMEF) for OC predictions. Plotted regions include Northeast (NE), Midwest (MW), Southeast (SE), Plains (PL), Northwest (NW), Southwest (SW) and total (T) and are defined as visualized in Fig. 8.







**Figure 11.** Spatial distribution of products form the LEBR CONUS11 simulation. Products include OA concentration (row 1), fraction of POA (row 2), pcSOA concentration (row 3) and change in model predicted POA from the nonvolatile POA model to the semivolatile POA model (row 4). Maps illustrate the median of all annual surface data (left), winter months (December, January, February; middle) and summer months (June, July, August; right).



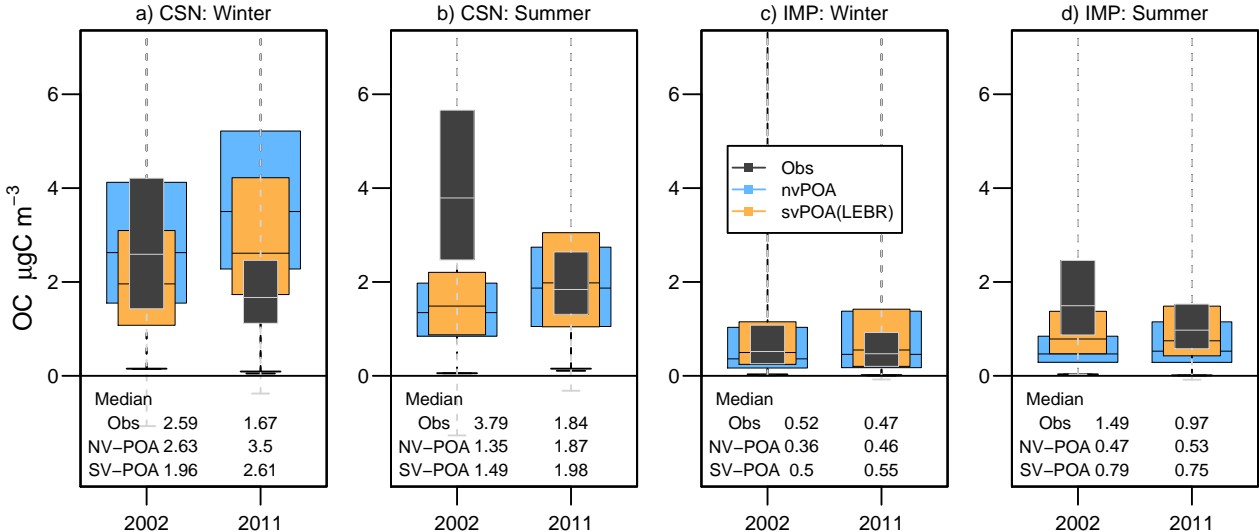

**Figure 12.** Organic carbon (OC) distributions for routine monitoring stations and model predictions at the same locations and times. The meaning of the boxes and whiskers is explained in Fig. 5.

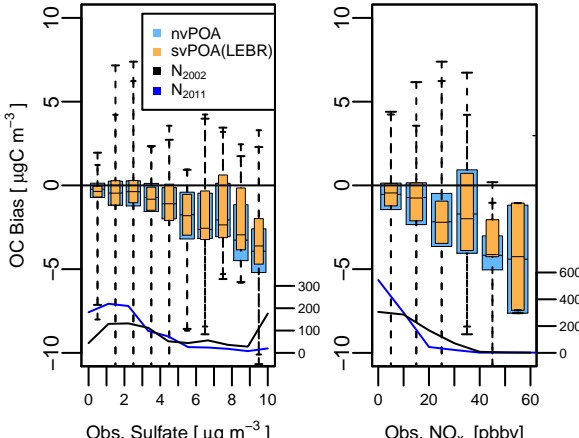

**Figure 13.** Organic carbon bias for CSN summer time data during 2002 and 2011 (combined) as a function of observed sulfate concentrations (left) and observed $NO_x$ concentrations (right). The meaning of boxes and whiskers is explained in Fig. 5. Blue (2011) and black (2002) solid lines quantify the number of data points for each year as functions of the observed pollutants.