# Peer review of "Semivolatile POA and parameterized total combustion SOA in CMAQv5.2: impacts on source strength and partitioning"

_Atmospheric Chemistry and Physics, 2017_

## Referee Comment (RC1) · Anonymous Referee #1 · 11 May 2017

In this work, Murphy et al. present the v5.2 of the Community Multiscale Air Quality (CMAQ) model, which takes into account the semivolatile phase partitioning and photochemical aging of POA emissions, and its evaluation over multiple seasons and locations. In addition, they have introduced into the model a new surrogate species that represents the potential SOA from combustion emissions (pcSOA) and explore its sensitivity to the emission scaling factor and the oxidation rate constant. Overall, the manuscript is very well written and the presentation is clear. However, I have a major concern regarding the implementation of pcSOA and the way that is presented. I recommend this study for publication after considering the following comments.

General comments:

1. The authors implemented pcSOA into CMAQv5.2 by using the SIMPLE parameterization for SOA formation in polluted urban regions Hodzic and Jimenez (2011). The optimal empirical parameters used for the formation of pcSOA (emission factors and oxidation reaction rate constants) are based on the observed proportionality of SOA concentrations and excess CO downwind of urban locations (i.e., Mexico City and Pasadena) under low biomass burning conditions. Therefore, pcVOC emissions includes all the anthropogenically emitted organic vapors and pcSOA expresses all the anthropogenic SOA. However, CMAQ already includes emissions of traditional anthropogenic VOCs and emissions of SVOCs (e.g., VSVPO3) and IVOCs (i.e., VIVPO1) in the gas phase, and therefore, the formation of SOA from anthropogenic VOCs, IVOCs, and SVOCs is currently double counted.

2. The first author of the manuscript have proposed a very effective nomenclature for atmospheric organic aerosols (Murphy et al., 2014). I would expect some consistency and I suggest to the authors to follow the naming convention of Murphy et al. (2014). In fact, the progress in the field of organic aerosols has come along with many complex and inconsistent abbreviates, therefore, adopting a standardized nomenclature will benefit the effective communication of the results to the scientific community. However, this is only a stylistic preference and it is up to the authors to adopt it.

Specific comments:

1. I find the introduction section very well written. Please consider including the following recent articles as well: Page 2 lines 24-25: (Shiraiwa et al., 2017); Page 3 line 10: (Ots et al., 2016; Jathar et al., 2017); Page 3 lines 19-20: (Ma et al., 2016); Page 4 line 1: (Tsimpidi et al., 2017)

2. Page 5 line 25: The Robinson et al. (2007) distribution includes emission factors up to the 106 $\mu$g m-3 volatility bin. However, CMAQv5.2 only have species with saturation concentration up to 104 $\mu$g m-3. Can you please include into the manuscript (maybe in Table 1?) the emission factors used for each of the primary organic material?

3. Page 6 lines 10-15: Does the oxidation of SVOC result in products with both higher and lower volatility than the precursor? I would suggest adding a reaction as an example and reporting the stoichiometric coefficients that you are using for each of the products.

4. Section 2.2: Do VOCs participate to only one oxidation step or are subject to multi-generational aging? Either way, I believe that SOA from anthropogenic VOCs is double counted due to the presence of pcSOA (see above in the first general comment). Hodzic and Jimenez (2011) do not include any traditional anthropogenic VOC specie (e.g., benzene) other than the lumped surrogate anthropogenic specie VOCA, which is analogous to the pcVOC used here.

5. Page 7 line 3: I am not sure that the name "pcVOC" is the more appropriate to use since this specie also includes unspeciated SVOC/IVOC emissions.

6. Page 7 line 25: The saturation concentration reported here (10-3 $\mu$g m-3) is not the same with the value reported in Table 2 (10-5 $\mu$g m-3).

7. Page 11 line 32: The sum of the fractions reported here is 99%.

8. Page 12 line 4: Please add the fractional biases of your nonvolatile POA simulation for comparison.

9. Page 19 line 31: Please consider deleting the "from anthropogenic and biogenic carbon sources".

10. Figures 1, 2 and 3: Please add in the figures captions what the boxes and the whiskers represent

11. Figure 2: According to the figure caption, the observed values over Birmingham, Atlanta and Yorkville are organic carbon (in $\mu$gC m-3). Are the modelled values converted to OC as well? I would suggest converting everything (both observed and modelled values) to OA (in $\mu$g m-3) in order to be consistent with the values over Centreville and Look Rock, which are shown in the same figure, as well as with the title of the y

axis.

12. Figure 3. Please add here (and discuss in the text) the model performance for POA and OOA over Centreville, Look Rock and Sacramento where there are also AMS measurements available.

13. Figure 5: I would recommend deleting the boxes and the whiskers behind each trend and keeping only the lines. This figure is too overloaded and it is very difficult for the reader to take any extra-information other than the trends of the lines.

References

Hodzic, A. and Jimenez, J. L.: Modeling anthropogenically controlled secondary organic aerosols in a megacity: a simplified framework for global and climate models, Geosci. Model Dev., 4, 901-917, 2011.

Jathar, S. H., Woody, M., Pye, H. O. T., Baker, K. R., and Robinson, A. L.: Chemical transport model simulations of organic aerosol in southern California: model evaluation and gasoline and diesel source contributions, Atmos. Chem. Phys., 17, 4305-4318, 2017.

Ma, P. K., Zhao, Y., Robinson, A. L., Worton, D. R., Goldstein, A. H., Ortega, A. M., Jimenez, J. L., Zotter, P., Prévôt, A. S. H., Szidat, S., and Hayes, P. L.: Evaluating the impact of new observational constraints on P-S/IVOC emissions, multi-generation oxidation, and chamber wall losses on SOA modeling for Los Angeles, CA, Atmos. Chem. Phys. Discuss., 1-35, 2016.

Murphy, B. N., Donahue, N. M., Robinson, A. L., and Pandis, S. N.: A naming convention for atmospheric organic aerosol, Atmos. Chem. Phys., 14, 5825-5839, 2014.

Ots, R., Young, D. E., Vieno, M., Xu, L., Dunmore, R. E., Allan, J. D., Coe, H., Williams, L. R., Herndon, S. C., Ng, N. L., Hamilton, J. F., Bergström, R., Di Marco, C., Nemitz, E., Mackenzie, I. A., Kuenen, J. J. P., Green, D. C., Reis, S., and Heal, M. R.: Simulating secondary organic aerosol from missing diesel-related intermediate-volatility organic

compound emissions during the Clean Air for London (ClearfLo) campaign, Atmos. Chem. Phys., 16, 6453-6473, 2016.

Robinson, A. L., Donahue, N. M., Shrivastava, M. K., Weitkamp, E. A., Sage, A. M., Grieshop, A. P., Lane, T. E., Pierce, J. R., and Pandis, S. N.: Rethinking organic aerosols: Semivolatile emissions and photochemical aging, Science, 315, 1259-1262, 2007.

Shiraiwa, M., Li, Y., Tsimpidi, A. P., Karydis, V. A., Berkemeier, T., Pandis, S. N., Lelieveld, J., Koop, T., and Poschl, U.: Global distribution of particle phase state in atmospheric secondary organic aerosols, Nature Communications, 8, 2017.

Tsimpidi, A. P., Karydis, V. A., Pandis, S. N., and Lelieveld, J.: Global-scale combustion sources of organic aerosols: Sensitivity to formation and removal mechanisms, Atmos. Chem. Phys. Discuss., 2017, 1-38, 2017.

---

## Referee Comment (RC2) · Anonymous Referee #2 · 26 May 2017

The manuscript by Murphy et al. reports on the revised treatment of organic aerosol (OA) in the Community Multiscale Air Quality Model (CMAQ) v5.2. The revised treatment of OA includes: 1. partitioning and gas-phase aging of primary OA (POA), and 2. a new model species "pc" (pcVOA, pcSOG, pcSOA) that represents the missing emissions and processes that may be associated with secondary OA (SOA) formation from urban combustion sources. Model simulations are performed at 4- and 12-km resolution and evaluated at different time periods, seasons, and US locations using surface network monitor data. The changes in the model representation of OA generally result in better correlation and improved bias; the average contribution of the new model species, pcSOA, to OA was ~39% in winter and ~24% in summer. The model evaluation is thorough and may contribute to elucidating the relative importance of emissions vs. processing in specific locations. The manuscript is generally well written and appropriate for publication in ACP. Specific comments and suggestions for revision are provided below.

In the abstract and introduction the authors suggest that given the quality of the model predictions using the "simple" parameterization presented, caution should be exercised when using more complicated parameterizations (higher number of uncertain parameters). I do not see a fundamental difference with the approach presented here and others. SOA formation depends on the amount of precursor, the extent of oxidation, and volatility of the resultant oxidation/reaction products. The approach presented is a hybrid of existing approaches (including VBS for POA) and relies on arguably uncertain parameters for each of the factors controlling SOA from the new precursor (scaling factor for POA to determine pcVOC, reaction rate constant with kOH to obtain pcSOG, and c* value to convert pcSOG to pcSOA). The net result of all of the current modeling approaches is that relative to the traditional two-product/non-volatile POA approach, they produce more oxidized OA with a temporal and spatial distribution that is more representative of observations. This is not to say that the changes in the model representation aren't warranted or needed; they are. As articulated by the authors, the changes represent the evolving knowledge of OA formation in the atmosphere. However, all of the current approaches face the same limitations regarding uncertainty in model parameters, a consequence of the complexity and likely variability of processes contributing to SOA formation.

The approach presented combines traditional model representation for VOCs, VBS model representation for POA, and a method based on Hodzic and Jimenez to represent missing sources and processes. It would be useful to see the relative contributions of these processes (by model species) to the total OA predicted. Were any simulations run with only the partitioning and aging of POA or only the consideration of pcSOA? On p. 5, line 34 the authors state that no POA emissions scaling was used to introduce

SVOCs; however, that is effectively what is done using the pc surrogate species. If combustion source emissions inventories have been revised to reflect the knowledge of dynamic partitioning and missing IVOCs, then these emissions are being double counted by the combined use of a dynamic POA model and the scaling of pcVOC to POA emission rate.

What is the rationale for maintaining the SOA formation pathway for traditional VOC precursors? Isn't this pathway essentially accounted for (or could be accounted for) using the surrogate?

It is recommended that the products of POA evaporation and aging ("OO") be listed in section 2.1, similarly to the directly emitted species, to improve the clarity of Table 1. How are the molecular weights assigned to the "OO" products?

On page 7, line 25 the authors note that no further reactions are considered for pc-SOG/pcSOA, but based on the very low c* and the rationale for including pcSOA, doesn't the conversion of pcSOG to pcSOA essentially represent these "other" reactions? Given that all of the pc is likely to end up (and stay) in the aerosol phase, it isn't clear what other reactions would be considered or why.

It is suggested that the authors use naming conventions that have been presented previously. For example, "LO-OOA" is presented on pg 9, line 11. Is the same as AMS derived "OOA-II"?

On page 15, line 14: What is meant by wood burning area sources not emitting SOA precursors consistent with pcSOA formation?

The value of c* for pc is 10-3 in the text and 10-5 in table 2.
* * *

---

## Author Comment (AC1) · 19 Jul 2017

**Responses to Referee #1**

In this work, Murphy et al. present the v5.2 of the Community Multiscale Air Quality (CMAQ) model, which takes into account the semivolatile phase partitioning and photochemical aging of POA emissions, and its evaluation over multiple seasons and locations. In addition, they have introduced into the model a new surrogate species that represents the potential SOA from combustion emissions (pcSOA) and explore its sensitivity to the emission scaling factor and the oxidation rate constant. Overall, the manuscript is very well written and the presentation is clear. However, I have a major concern regarding the implementation of pcSOA and the way that is presented. I recommend this study for publication after considering the following comments.

We thank the reviewer for their careful consideration of our manuscript and sharing constructive feedback. We have accepted their revisions in almost every case and are grateful for the added clarity they will provide. Where we disagree with the reviewer, we have explained our rationale fully in the responses below, and we look forward to continued dialogue.

General comments:

1. The authors implemented pcSOA into CMAQv5.2 by using the SIMPLE parameterization for SOA formation in polluted urban regions Hodzic and Jimenez (2011). The optimal empirical parameters used for the formation of pcSOA (emission factors and oxidation reaction rate constants) are based on the observed proportionality of SOA concentrations and excess CO downwind of urban locations (i.e., Mexico City and Pasadena) under low biomass burning conditions. Therefore, pcVOC emissions includes all the anthropogenically emitted organic vapors and pcSOA expresses all the anthropogenic SOA. However, CMAQ already includes emissions of traditional anthropogenic VOCs and emissions of SVOCs (e.g., VSVPO3) and IVOCs (i.e., VIVPO1) in the gas phase, and therefore, the formation of SOA from anthropogenic VOCs, IVOCs, and SVOCs is currently double counted.

The reviewer is correct in their description of the SIMPLE parameterization and its basis in proportionality of SOA to excess CO. However, the parameters we have chosen for this study do not correspond exactly to those used previously. First, the emissions of the VOC precursor in Hodzic and Jimenez (2011) were scaled to CO emissions; in this study, they are scaled to POA emissions. For our base case scaling factor, we began with the optimal parameter from Hayes et al. (2013), 0.069 g VOC (g CO)$^{-1}$, and we converted it to g VOC (g POA)$^{-1}$ using the average ratio of CO to POA concentrations predicted by the model in the California domain. For the California scenarios, this combination of parameters worked well. For the larger domain, as we show in the manuscript, the existing measurements suggested that we decrease the emission scale factor by 32%. The take-home point is that we are not applying the parameters from Mexico City and Pasadena studies a priori to our cases. Instead, we are using a similar method after accounting for all of the known sources that CMAQ has included in the past. In this way, pcSOA only accounts for potentially missing sources and pathways.

2. The first author of the manuscript have proposed a very effective nomenclature for atmospheric organic aerosols (Murphy et al., 2014). I would expect some consistency and I suggest to the authors to follow the naming convention of Murphy et al. (2014). In fact, the progress in the field of organic aerosols has come along with many complex and inconsistent abbreviates, therefore, adopting a standardized nomenclature will benefit the effective communication of the results to

the scientific community. However, this is only a stylistic preference and it is up to the authors to adopt it.

We appreciate the reviewer's suggestion and did of course use that previous publication as the starting point for our choice of names in the current scheme. As it stands, there are pre-existing constraints in CMAQ for naming chemical species (e.g. all particle-phase species begin with "A", aqueous SOA compounds already existed in the model, biomass burning was not considered separately here, etc). Additionally, we are not making use of an important component of the Murphy et al. (2014) scheme, the tracking of volatility at the point of emission. In future versions of the model, we anticipate that more elements of that scheme will become accessible to us and help standardize the nomenclature to a greater extent.

Specific comments:

1. I find the introduction section very well written. Please consider including the following recent articles as well: Page 2 lines 24-25: (Shiraiwa et al., 2017); Page 3 line 10: (Ots et al., 2016; Jathar et al., 2017); Page 3 lines 19-20: (Ma et al., 2016); Page 4 line 1: (Tsimpidi et al., 2017).

The references have been added to the manuscript as the reviewer suggests.

2. Page 5 line 25: The Robinson et al. (2007) distribution includes emission factors up to the $10^6$ µg m$^{-3}$ volatility bin. However, CMAQv5.2 only have species with saturation concentration up to $10^4$ µg m$^{-3}$. Can you please include into the manuscript (maybe in Table 1?) the emission factors used for each of the primary organic material?

The emission factors have been added to table 1.

**Table 1.** Properties of semivolatile POA species in CMAQv5.2.

| Species Name (Particle/Vapor) | Mol. Weight g mol$^{-1}$ | $C^{*,1}$ µg m$^{-3}$ | $N_C$ | O:C | $H_{vap}$ kJ mol$^{-1}$ | POA Emission Frac.[2] |
|---|---|---|---|---|---|---|
| ALVPO1/VLVPO1[2,3] | 218 | 0.1 | 13.0 | 0.185 | 89 | 0.09 |
| ASVPO1/VSVPO1[3,4] | 230 | 1 | 14.5 | 0.123 | 85 | 0.09 |
| ASVPO2/VSVPO2 | 241 | 10 | 16.0 | 0.073 | 81 | 0.14 |
| ASVPO3/VSVPO3 | 253 | 100 | 17.5 | 0.032 | 77 | 0.18 |
| AIVPO1/VIVPO1[4,5] | 266 | 1000 | 19.0 | 0.00 | 73 | 0.5 |
| ALVOO1/VLVOO1[5,6] | 136 | 0.01 | 5.0 | 0.886 | 93 | NA |
| ALVOO2/VLVOO2 | 136 | 0.1 | 5.5 | 0.711 | 89 | NA |
| ASVOO1/VSVOO1 | 135 | 1 | 6.0 | 0.567 | 85 | NA |
| ASVOO2/VSVOO2 | 135 | 10 | 6.5 | 0.447 | 81 | NA |
| ASVOO3/VSVOO3 | 134 | 100 | 7.0 | 0.345 | 77 | NA |

[1] $C^*$ values are defined at reference temperature 298 K.
[2] Robinson et al. (2007)
[3] A = Aerosol; V = Vapor; LV = Low Volatility; PO = Primary Organic
[4] SV = Semivolatile
[5] IV = Intermediate Volatility
[6] OO = Oxidized Organic

3. Page 6 lines 10-15: Does the oxidation of SVOC result in products with both higher and lower volatility than the precursor? I would suggest adding a reaction as an example and reporting the stoichiometric coefficients that you are using for each of the products.

Yes, each reaction takes into account fragmentation consistent with Eq. 1. Thus products of higher and lower volatility are produced. We have added an example reaction and its stoichiometry to the supplemental information. Further, we have added the entire stoichiometry for the aging mechanism to the supplemental information.

"Table S1 shows the stoichiometric coefficients derived from the branching ratio formula (Eq. 1) and the aging kernel published in Donahue et al. (2012). As an example, we provide the oxidation stoichiometry of the oxidation of VSVPO2, one of the surrogates for primary semivolatile vapors:

$$VSVPO2 + OH \rightarrow OH + 0.3856 * VLVPO1 + 0.0950 * VSVPO1 \qquad Eq. S2$$
$$+ 0.1373 * VSVPO2 + 0.0005 * VSVPO3$$
$$+ 0.2051 * VLVOO1 + 0.1764 * VLVOO2$$

These molar yields are derived in order to conserve carbon. Hydroxyl radicals are assumed to be regenerated by the oxidation reactions so that the aging of POA does not perturb the oxidant budget; the overall reactivity associated with POA mass is small compared to that associated with VOCs. The stoichiometric coefficients indicate that low O:C surrogates (VLVPO1-VIVPO1) in addition to high O:C surrogates (VLVOO1-VSVOO3). This feature ensures that the evolution of bulk O:C behaves similarly to the more detailed approaches. This is described in more detail in the following section."

4. Section 2.2: Do VOCs participate to only one oxidation step or are subject to multigenerational aging? Either way, I believe that SOA from anthropogenic VOCs is double counted due to the presence of pcSOA (see above in the first general comment). Hodzic and Jimenez (2011) do not include any traditional anthropogenic VOC species (e.g., benzene) other than the lumped surrogate anthropogenic specie VOCA, which is analogous to the pcVOC used here.

The secondary compounds formed from VOC oxidation are subject to only the first oxidation step (i.e. no multigenerational aging) consistent with Griffin et al. (1999). We acknowledge the reviewer's point and concede that it is possible that some of the anthropogenic SOA from multigenerational aging of VOCs is "double-counted" by the model configuration. However, the total first-generation contribution of SOA from anthropogenic VOCs predicted by the model at Pasadena, for example, is minor (9.0%) compared to the contribution of pcSOA predicted (67.6%). We have added a speciated plot depicting the average contributions of individual OA categories to the total in the supporting information (Fig. S4). This simple comparison illustrates the significant gap between the first-generation anthropogenic SOA yields in CMAQ and observations of urban SOA dominated by vehicles as well as other sources. Although we have taken great care in this study to recommend a set of robust parameters for pcSOA formation, this assessment relies on data from specific time periods, like 2011 over the continental U.S. We emphasize that these parameters will be revised in the future as better-constrained parameters for anthropogenic OA formation, like multigenerational SOA yields, volatility-resolved emission factors, and heterogeneous reaction rates, etc, are implemented. These improvements will enable

a more consistent bottom-up understanding of the urban OA problem, but until then, this top-down approach provides useful predictions of bulk OA formation and exposure.

5. Page 7 line 3: I am not sure that the name "pcVOC" is the more appropriate to use since this species also includes unspeciated SVOC/IVOC emissions.

Although the reviewer's point is well-taken, we prefer to keep the existing name for this surrogate species. We consider the term VOC as it used here to be a more appropriate term based on how it is implemented in the model, because partitioning of this species, even in very cold (e.g. upper troposphere) or very concentrated (e.g. biomass burning plumes) conditions is not considered. We agree with the reviewer about the possibility of confusion here, so we have added a statement to the methods section 2.3.

"A new surrogate VOC species (potential VOC from combustion emissions, pcVOC) is introduced with an emission rate that is scaled to the POA mass emission rate. This species does not partition directly to the particle phase and in that respect, behaves as a VOC in the model. It:is oxidized with OH to form a low volatility condensable vapor, potential secondary organic gas from combustion emissions, pcSOG (table 2)."

6. Page 7 line 25: The saturation concentration reported here ($10^{-3}$ _g m-3) is not the same with the value reported in Table 2 ($10^{-5}$ _g m-3).

Updated.

7. Page 11 line 32: The sum of the fractions reported here is 99%.

The reported percentages in the original study sum to 99% rather than 100% due to rounding error. Please see Liu et al. (2012).

8. Page 12 line 4: Please add the fractional biases of your nonvolatile POA simulation for comparison.

Updated.

9. Page 19 line 31: Please consider deleting the "from anthropogenic and biogenic carbon sources".

We have made the change.

10. Figures 1, 2 and 3: Please add in the figures captions what the boxes and the whiskers represent.

We have added this information to the Figure captions.

"The boxes denote the 25th and 75th percentiles of each dataset while the whiskers extend to the most extreme points."

11. Figure 2: According to the figure caption, the observed values over Birmingham, Atlanta and Yorkville are organic carbon (in µgC m$^{-3}$). Are the modelled values converted to OC as well? I would suggest converting everything (both observed and modelled values) to OA (in µg m$^{-3}$) in

order to be consistent with the values over Centreville and Look Rock, which are shown in the same figure, as well as with the title of the y axis.

The model values are in µgC m⁻³ for Birmingham, Atlanta, and Yorkville. Because we view the application of OM:OC ratios from the model to be more appropriate than using uncertain OM:OC ratios for the field data, we prefer to keep the comparisons that are in OC units. We also point out that the comparison across sites is not the focus of this analysis. Instead, we seek the most reliable comparison of modeled to measured data at each site independently. We agree that the y-axis is confusing, and we have updated the figure to be more clear on this point.

[Figure]

12. Figure 3. Please add here (and discuss in the text) the model performance for POA and OOA over Centreville, Look Rock and Sacramento where there are also AMS measurements available.

Unfortunately, HOA/OOA factors for these three sites are not available. The PMF analyses for Sacramento does not exist to our knowledge. The analyses at Centreville and Look Rock were able to isolate a small number of factors including an aged OA, an isoprene-derived OA and a biomass burning OA factor. However, the relative contributions of these factors would not have been affected significantly by the improvements made in this study, which are related to the anthropogenic combustion OA. Displaying these factors in Fig. 3 would also not be appropriate since none of them correspond to the HOA factor already present. We have added a statement to the manuscript explaining why these comparisons are absent.

> "Figure 3 compares CMAQ OA species to AMS factors derived from PMF analysis at Pasadena, Bakersfield, and Cool. PMF analysis was not available at the Sacramento site, and the Centreville and Look Rock sites showed negligible presence of HOA throughout the SOAS campaign."

13. Figure 5: I would recommend deleting the boxes and the whiskers behind each trend and keeping only the lines. This figure is too overloaded and it is very difficult for the reader to take any extra-information other than the trends of the lines.

Updated

References

Hodzic, A. and Jimenez, J. L.: Modeling anthropogenically controlled secondary organic aerosols in a megacity: a simplified framework for global and climate models, Geosci. Model Dev., 4, 901-917, 2011.

Jathar, S. H., Woody, M., Pye, H. O. T., Baker, K. R., and Robinson, A. L.: Chemical transport model simulations of organic aerosol in southern California: model evaluation and gasoline and diesel source contributions, Atmos. Chem. Phys., 17, 4305-4318, 2017.

Ma, P. K., Zhao, Y., Robinson, A. L., Worton, D. R., Goldstein, A. H., Ortega, A. M., Jimenez, J. L., Zotter, P., Prévôt, A. S. H., Szidat, S., and Hayes, P. L.: Evaluating the impact of new observational constraints on P-S/IVOC emissions, multi-generation oxidation, and chamber wall losses on SOA modeling for Los Angeles, CA, Atmos. Chem. Phys. Discuss., 1-35, 2016.

Murphy, B. N., Donahue, N. M., Robinson, A. L., and Pandis, S. N.: A naming convention for atmospheric organic aerosol, Atmos. Chem. Phys., 14, 5825-5839, 2014.

Ots, R., Young, D. E., Vieno, M., Xu, L., Dunmore, R. E., Allan, J. D., Coe, H., Williams, L. R., Herndon, S. C., Ng, N. L., Hamilton, J. F., Bergström, R., Di Marco, C., Nemitz, E., Mackenzie, I. A., Kuenen, J. J. P., Green, D. C., Reis, S., and Heal, M. R.: Simulating secondary organic aerosol from missing diesel-related intermediate-volatility organic compound emissions during the Clean Air for London (ClearfLo) campaign, Atmos. Chem. Phys., 16, 6453-6473, 2016.

Robinson, A. L., Donahue, N. M., Shrivastava, M. K., Weitkamp, E. A., Sage, A. M., Grieshop, A. P., Lane, T. E., Pierce, J. R., and Pandis, S. N.: Rethinking organic aerosols: Semivolatile emissions and photochemical aging, Science, 315, 1259-1262, 2007.

Shiraiwa, M., Li, Y., Tsimpidi, A. P., Karydis, V. A., Berkemeier, T., Pandis, S. N., Lelieveld, J., Koop, T., and Poschl, U.: Global distribution of particle phase state in atmospheric secondary organic aerosols, Nature Communications, 8, 2017.

Tsimpidi, A. P., Karydis, V. A., Pandis, S. N., and Lelieveld, J.: Global-scale combustion sources of organic aerosols: Sensitivity to formation and removal mechanisms, Atmos. Chem. Phys. Discuss., 2017, 1-38, 2017.

References for Response

Griffin, R. J., et al. (1999). "Estimate of global atmospheric organic aerosol from oxidation of biogenic hydrocarbons." Geophysical Research Letters **26**(17): 2721-2724.

Hodzic, A. and Jimenez, J. L.: Modeling anthropogenically controlled secondary organic aerosols in a megacity: a simplified framework for global and climate models, Geosci. Model Dev., 4, 901-917, 2011.

Murphy, B. N., Donahue, N. M., Robinson, A. L., and Pandis, S. N.: A naming convention for atmospheric organic aerosol, Atmos. Chem. Phys., 14, 5825-5839, 2014.

---

## Author Comment (AC2)

**Responses to Referee #2**

(Please note that a revised version of the manuscript with changes tracked and the supporting information appear below)

The manuscript by Murphy et al. reports on the revised treatment of organic aerosol (OA) in the Community Multiscale Air Quality Model (CMAQ) v5.2. The revised treatment of OA includes: 1. partitioning and gas-phase aging of primary OA (POA), and 2. a new model species "pc" (pcVOA, pcSOG, pcSOA) that represents the missing emissions and processes that may be associated with secondary OA (SOA) formation from urban combustion sources. Model simulations are performed at 4- and 12-km resolution and evaluated at different time periods, seasons, and US locations using surface network monitor data. The changes in the model representation of OA generally result in better correlation and improved bias; the average contribution of the new model species, pcSOA, to OA was _39% in winter and _24% in summer. The model evaluation is thorough and may contribute to elucidating the relative importance of emissions vs. processing in specific locations. The manuscript is generally well written and appropriate for publication in ACP. Specific comments and suggestions for revision are provided below.

We thank the reviewer for helpful suggestions in improving the depth and consistency of the manuscript. We have provided detailed responses to each individual point below.

In the abstract and introduction, the authors suggest that given the quality of the model predictions using the "simple" parameterization presented, caution should be exercised when using more complicated parameterizations (higher number of uncertain parameters). I do not see a fundamental difference with the approach presented here and others. SOA formation depends on the amount of precursor, the extent of oxidation, and volatility of the resultant oxidation/reaction products. The approach presented is a hybrid of existing approaches (including VBS for POA) and relies on arguably uncertain parameters for each of the factors controlling SOA from the new precursor (scaling factor for POA to determine pcVOC, reaction rate constant with kOH to obtain pcSOG, and c* value to convert pcSOG to pcSOA). The net result of all of the current modeling approaches is that relative to the traditional two-product/non-volatile POA approach, they produce more oxidized OA with a temporal and spatial distribution that is more representative of observations. This is not to say that the changes in the model representation aren't warranted or needed; they are. As articulated by the authors, the changes represent the evolving knowledge of OA formation in the atmosphere. However, all of the current approaches face the same limitations regarding uncertainty in model parameters, a consequence of the complexity and likely variability of processes contributing to SOA formation.

We agree with the reviewer that there are uncertain parameters involved in each of the factors controlling SOA formation in the pcSOA approach. Our statement cautioning readers regarding complicated parameterizations was not meant to distinguish our approach from others in the literature or communicate superiority but rather to point out explicitly that our results indicate a model does not need an overly complicated approach in order to achieve arguably good performance when compared with existing configurations. This performance is not only satisfactory in urban areas where the simple approach has been parameterized before, but is also substantially useful for regional-scale predictions at routine monitoring sites. Although much can be learned from the complex approaches, including source attribution, phase distribution, and chemical evolution, it is very difficult to evaluate these findings with independent observations. For a policy-driven model like CMAQ that is used routinely to develop air quality policies, independent evaluation and justification is paramount. We look forward to improving the

representation of these important pathways in the future with bottom-up approaches, but with careful consideration of the balance between uncertainty and improvement to model skill, especially when potential improvements may dramatically affect source attribution.

The approach presented combines traditional model representation for VOCs, VBS model representation for POA, and a method based on Hodzic and Jimenez to represent missing sources and processes. It would be useful to see the relative contributions of these processes (by model species) to the total OA predicted. Were any simulations run with only the partitioning and aging of POA or only the consideration of pcSOA?

We have added figures to the supplement (Figs. S4 and S5) showing the separate contribution of pcSOA, POA, oxidation products of POA aging, SOA formed in the aqueous phase, and SOA from traditional VOC precursors to the total OA at each site. We have also rerun the model as the reviewer asks and added figures (Figs. S5-S7) showing the difference in OA predictions at the CONUS scale in January and July with the pcSOA pathway turned off.

[Figure]

**Figure S4.** Composition of organic aerosol predicted by CMAQv5.2 run for both nonvolatile POA (NV) and semivolatile POA with pcSOA (SV). The species depicted include POA with very low O:C (POA), POA with high O:C (oxygenated POA or Oxy. POA), potential combustion SOA (pcSOA), SOA from traditional anthropogenic and biogenic VOCs (V-SOA), and SOA formed in the aqueous aerosol and cloud water phases (AQ-SOA). The sites shown in this figure include Pasadena (PD), Bakersfield (BK), Sacramento (SM), Cool (CL), Centreville (CN), Look Rock (LR), Birmingham (BH), Atlanta (AT), and Yorkville (YK). The rightmost two comparisons show the average contributions to OA for grid cells in the continental US from the CONUS11 simulation during July (CS07) and January (CS01). The higher contribution of AQ-SOA at the sites in the southeast US is a result of that simulation including isoprene and terpene aqueous-phase formation pathways that are not present in the California and CONUS-scale simulations.

[Figure]

**Figure S5.** Composition of organic aerosol predicted by CMAQv5.2 run for both nonvolatile POA (NV), semivolatile POA with no pcSOA (NP), and semivolatile POA with pcSOA (SV). The species depicted include POA with very low O:C (POA), POA with high O:C (oxygenated POA or Oxy. POA), potential combustion SOA (pcSOA), SOA from traditional anthropogenic and biogenic VOCs (V-SOA), and SOA formed in the aqueous aerosol and cloud water phases (AQ-SOA). The concentrations are average contributions to OA for grid cells in the continental US from the CONUS11 simulation during July (CONUS-July) and January (CONUS-Jan).

[Figure]

**Figure S6.** Average OA concentrations for nonvolatile POA (nvPOA), semivolatile POA with no pcSOA (No PcSOA), and semivolatile POA with pcSOA (svPOA(LEBR)). This simulation is for July, 2011.

[Figure]

**Figure S7.** Average OA concentrations for nonvolatile POA (nvPOA), semivolatile POA with no pcSOA (No PcSOA), and semivolatile POA with pcSOA (svPOA(LEBR)). This simulation is for January, 2011.

On p. 5, line 34 the authors state that no POA emissions scaling was used to introduce SVOCs; however, that is effectively what is done using the pc surrogate species. If combustion source emissions inventories have been revised to reflect the knowledge of dynamic partitioning and missing IVOCs, then these emissions are being double counted by the combined use of a dynamic POA model and the scaling of pcVOC to POA emission rate.

Previous literature documenting implementation of POA volatility scaling address two quantities. First, the original POA emissions factor is distributed up to $C^* = 10^3$ or $10^4$ µg m$^{-3}$ depending on the model species available (Shrivastava et al., 2008; Murphy et al., 2009; Koo et al., 2014). When we discuss missing SVOCs and IVOCs, we are not referring to mass in this first quantity as, by definition, it was detected during characterization of the emission factors that informed the emission inventory. The second quantity addressed in the literature represents the missing SVOCs and IVOCs and is usually scaled between 150% and 750% of the original POA emission factor (Shrivastava et al. 2008, Matsui et al., 2014).

Currently, we are engaged in a parallel research effort to analyze the data in existing combustion inventories and estimate the amount of missing S- and IVOCs at a source level with emerging knowledge of source-specific volatility distributions. For the current study though, we choose not to add these emissions into the POA volatility-resolved species, and instead we have lumped them with the other uncertain pathways discussed in the manuscript in order to generate the emissions of pcVOC. We are not adding the missing IVOC emissions twice.

What is the rationale for maintaining the SOA formation pathway for traditional VOC precursors? Isn't this pathway essentially accounted for (or could be accounted for) using the surrogate?

The SOA formation pathways for traditional VOC precursors is maintained for compatibility with both past versions of CMAQ and as a benchmark for evaluating future improvements to bottom-up approaches. The traditional approach of applying SOA yields to the oxidation of specific precursors is a valuable one, especially for attributing bulk OA mass to specific emission sources, connecting laboratory experiments to ambient observations, and evaluating potential regulatory actions (e.g., fuel reformulation). This pathway could be partly accounted for with the surrogate, although there are some problems. First, there are evaporative, non-combustion, sources of some traditional VOCs that are also responsible for making SOA. Second, the oxidation products of

these VOC species are semivolatile and their yields are $NO_x$-dependent in the model, and these behaviors are important to take into account.

It is recommended that the products of POA evaporation and aging ("OO") be listed in section 2.1, similarly to the directly emitted species, to improve the clarity of Table 1. How are the molecular weights assigned to the "OO" products?

The products of POA aging are listed in section 2.1 as the reviewer requests (original manuscript page 5, line 18). The molecular weights are assigned assuming each set of compounds comprises only carbon, oxygen, and hydrogen atoms. The O:C is calculated from the carbon number and volatility using Donahue et al. (2011). We further employ the oft-used parameterization of Heald et al. (2010) to calculate H:C from O:C. We have included these details in the manuscript as well.

> "The molecular weights of these species are calculated using the given carbon number and OM:OC, while assuming each representative species comprises only carbon, oxygen and hydrogen atoms. The H:C is calculated from O:C using insights from Heald et al. (2010), a common assumption for deriving molecular weights of VBS species in chemical transport models."

> And to the Supporting information, we have added the following:

> "The volatility of each surrogate species is calculated as a function of its given $C^*$ and O:C. Specifically, we use the following relationship:

> $$Log_{10}C^* = 0.475 * ( 25 - n_C ) + 2.3 * n_O + 0.6 * n_C * n_O / ( n_C + n_O ) \qquad \text{Eq. S1}$$

> where $C^*$ is the saturation concentration of the surrogate species in µg m$^{-3}$, $n_C$ is the number of carbon atoms in the species, and $n_O$ is the number of oxygen atoms in the species (Donahue et al., 2012). "

On page 7, line 25 the authors note that no further reactions are considered for pc-SOG/pcSOA, but based on the very low c* and the rationale for including pcSOA, doesn't the conversion of pcSOG to pcSOA essentially represent these "other" reactions? Given that all of the pc is likely to end up (and stay) in the aerosol phase, it isn't clear what other reactions would be considered or why.

Some possible reactions would be particle-phase oxidation reactions or photolysis leading to degradation of the low volatility material (Hodzic et al., 2015). These reactions are not incorporated into CMAQ currently. If they were, they would increase the volatility of the OA species in the model and reduce aerosol-phase concentration. We have added these examples to the description of the methods.

> "Although heterogeneous reactions are implemented for other SOA types in CMAQ, no further reactions are included for pcSOA. Additionally, photolysis leading to degradation of low volatility material is not considered in the model (Hodzic et al., 2015)."

It is suggested that the authors use naming conventions that have been presented previously. For example, "LO-OOA" is presented on pg 9, line 11. Is the same as AMS derived "OOA-II"?

The "LO-OOA" term is used often by the AMS community. Zhang et al. (2011) suggested using this term instead of terms like LV-OOA and SV-OOA when volatility information is not available. Xu et al. (2016) also argue that the correlations between oxygenation as observed by the AMS

and volatility are highly variable among lab and field studies. To our understanding, the terms OOA-I and OOA-II are less often used now.

On page 15, line 14: What is meant by wood burning area sources not emitting SOA precursors consistent with pcSOA formation?

This refers back to the distinction made between fossil-fuel and biomass-burning emissions described in the methods section. Recent studies have shown limited downwind OA formation from biomass-burning sources, and the current model configuration takes that into account by not emitting pcVOC from wildfire sources. However, our emission inputs, as is the case for most CTMs include gridded area fires (e.g. residential wood burning, prescribed fires, etc) lumped together with vehicle and other fossil-fuel emissions. For this reason, there may be a significant discrepancy introduced in the comparison to wintertime measurements when gridded area biomass burning sources are important. We have revised the sentence.

> "Meanwhile in the wintertime cases, large wood burning area emissions may not result in substantial net OA formation downwind. Although the simulation results shown here take this possible feature into account for wildfire sources, residential and other smaller-scale wood combustion are assumed to produce pcSOA consistent with fossil-fuel sources. If these sources are significantly overpredicted, then lower pcSOA production rates would yield better agreement at the CONUS scale for the wrong reasons."

The value of $c^*$ for pc is 10-3 in the text and 10-5 in table 2.

Updated.

References for Response

Donahue, N. M., et al. (2011). "A two-dimensional volatility basis set: 1. organic-aerosol mixing thermodynamics." Atmospheric Chemistry and Physics **11**(7): 3303-3318.

Heald, C., et al. (2010). "A simplified description of the evolution of organic aerosol composition in the atmosphere." Geophysical Research Letters **37**(8).

Hodzic, A., et al. (2015). "Organic photolysis reactions in tropospheric aerosols: effect on secondary organic aerosol formation and lifetime." Atmos. Chem. Phys. **15**(16): 9253-9269.

Koo, B., et al. (2014). "1.5-Dimensional volatility basis set approach for modeling organic aerosol in CAMx and CMAQ." Atmospheric Environment **95**: 158-164.

Matsui, H., et al. (2014). "Volatility basis-set approach simulation of organic aerosol formation in East Asia: implications for anthropogenic–biogenic interaction and controllable amounts." Atmos. Chem. Phys. **14**(18): 9513-9535.

Murphy, B. N. and S. N. Pandis (2009). "Simulating the Formation of Semivolatile Primary and Secondary Organic Aerosol in a Regional Chemical Transport Model." Environmental Science & Technology **43**(13): 4722-4728.

Shrivastava, M. K., et al. (2008). "Effects of gas particle partitioning and aging of primary emissions on urban and regional organic aerosol concentrations." Journal of Geophysical Research **113**(D18).

Xu, L., Williams, L. R., Young, D. E., Allan, J. D., Coe, H., Massoli, P., Fortner, E., Chhabra, P., Herndon, S., Brooks, W. A., Jayne, J. T., Worsnop, D. R., Aiken, A. C., Liu, S., Gorkowski, K., Dubey, M. K., Fleming, Z. L., Visser, S., Prévôt, A. S. H., and Ng, N. L. (2016). Wintertime aerosol chemical composition, volatility, and spatial variability in the greater London area, Atmos. Chem. Phys., 16, 1139-1160, https://doi.org/10.5194/acp-16-1139-2016.

Zhang, Q., et al. (2011). "Understanding atmospheric organic aerosols via factor analysis of aerosol mass spectrometry: a review." Anal Bioanal Chem **401**(10): 3045-3067.